# DRoC: Elevating Large Language Models for Complex Vehicle Routing via Decomposed Retrieval of Constraints

**Xia Jiang**[1], **Yaoxin Wu**[1,*] **Chenhao Zhang**[2], **and Yingqian Zhang**[1]
[1]Eindhoven University of Technology, The Netherlands
[2]Southeast University, China
x.jiang1@tue.nl, y.wu2@tue.nl, zhangchenhaoseu@foxmail.com, yqzhang@tue.nl

## Abstract

This paper proposes Decomposed Retrieval of Constraints (DRoC), a novel framework aimed at enhancing large language models (LLMs) in exploiting solvers to tackle vehicle routing problems (VRPs) with intricate constraints. While LLMs have shown promise in solving simple VRPs, their potential in addressing complex VRP variants is still suppressed, due to the limited embedded internal knowledge that is required to accurately reflect diverse VRP constraints. Our approach mitigates the issue by integrating external knowledge via a novel retrieval-augmented generation (RAG) approach. More specifically, the DRoC decomposes VRP constraints, externally retrieves information relevant to each constraint, and synergistically combines internal and external knowledge to benefit the program generation for solving VRPs. The DRoC also allows LLMs to dynamically select between RAG and self-debugging mechanisms, thereby optimizing program generation without the need for additional training. Experiments across 48 VRP variants exhibit the superiority of DRoC, with significant improvements in the accuracy rate and runtime error rate delivered by the generated programs. The DRoC framework has the potential to elevate LLM performance in complex optimization tasks, fostering the applicability of LLMs in industries such as transportation and logistics.

## 1 Introduction

Vehicle Routing Problems (VRPs) are central to computer science and combinatorial optimization, serving as a foundational benchmark for evaluating computational complexity and developing approximation algorithms (Pillai & Singh, 2023). Beyond their theoretical significance, VRPs are also a major focus in operations research (OR), where they model decision-making processes in various industrial domains. The NP-hard challenge of VRPs in practice escalates substantially with the complexity of composite constraints, reflecting the interplay between theoretical and practical aspects. Different solvers such as OR-tools and Gurobi are commonly used due to their accessibility and generic modeling capabilities. Despite easy applications in simple VRPs, for users who lack modelling and optimization skills, these solvers are hard to use for solving VRPs with composite constraints, since 1) there are few example codes/documentation to explain the modeling of various constraints, and 2) developing programs for complex VRPs necessitates expert-level domain knowledge. Hence, it is challenging for non-experts to successfully apply the solvers to real-world operations (AhmadiTeshnizi et al., 2024). Consequently, researchers have increasingly focused on automating problem-solving procedures to mitigate dependence on domain and modelling expertise.

Large language models (LLMs) have demonstrated expert-level performance in several domains (Almeida et al., 2024) and have recently been applied to optimization problems in OR (Xiao et al., 2023; Zhang et al., 2024a). Their advanced reasoning and generation capabilities offer the potential to automate modeling and programming tasks. Despite the success in solving simple optimization problems, LLMs frequently face limitations when dealing with VRPs characterized by composite constraints (see Figure 1, which benchmarks GPT-3.5-turbo on 48 VRPs used in this paper). This

---

*Corresponding author

challenge arises from LLMs' bounded internal knowledge since the domain-specific corpus is insufficient during training processes. As a result, LLMs exhibit deficiencies in generating programs for VRPs, and they lack capabilities of 1) the accurate formulation of some specific constraints, and 2) the integration of heterogeneous constraints within a generated program. They pose significant obstacles to the widespread application of LLMs in solving complex VRPs in real-world scenarios, particularly those distinguished by intricate constraints. For instance, state-of-the-art (SOTA) LLM-based methods often fail to address complex problems due to incorrect constraint modeling with coding errors (AhmadiTeshnizi et al., 2024). Therefore, we aim for the integration of external knowledge into LLMs and target at improving constraint modeling in program generation for VRPs.

Inspired by Chain-of-Thought (CoT) (Wei et al., 2022) and Divide-and-Conquer (DaC) paradigms (Zhang et al., 2024b), which showcase that complex tasks can be solved by an LLM through a decomposed manner, we propose a systematic integration of external knowledge and decomposition techniques to enhance LLMs in program generation for VRP solvers. Specifically, we introduce a novel retrieval-augmented generation (RAG) framework, termed Decomposed Retrieval of Constraint (DRoC), which enables LLMs to more effectively address complex VRPs without additional training. The DRoC framework facilitates the incorporation of external knowledge retrieved from documentation and example codes. Notably, we perform constraint-based decomposition for the target VRP during the retrieval process, which further enhances the correctness and constraint-specificity of generated programs. In addi-

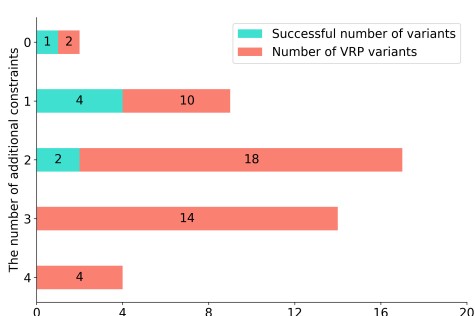

Figure 1: The evaluation of gpt-3.5-turbo on 48 VRP variants with different numbers of composite constraints. Performance declines with increased constraints.

tion, our framework synergistically combines external and internal knowledge by empowering LLMs to dynamically select between RAG and self-debugging mechanisms, continuously optimizing the program generation process. We conducted experiments across a set of 48 assorted VRPs and further extend the method to other OR problems, demonstrating the efficacy of the DRoC framework.

## 2 RELATED WORK

### 2.1 LLMs FOR VRPs

The advent of LLMs has facilitated advanced approaches to VRPs. LLMs can embed different problems by natural language and thereby enable a multi-task model for tackling simple OR problems, including basic travelling salesman problem (TSP) and capacitated vehicle routing problem (CVRP) (Jiang et al., 2024). The heuristics for addressing VRPs are automatically searched through LLMs with the aid of evolutionary computation (EC) (Liu et al., 2024; Ye et al., 2024). However, these methods typically aim to evolve pre-defined algorithm types such as guided local search, necessitating much domain-specific knowledge and prerequisites. Also, they often entail a substantial number of LLM invocations for evolution, e.g., for creating and maintaining a population of algorithms.

Alternative research focuses on the modeling and programming of OR problems including VRPs based on the textual descriptions. These approaches aim to transform user queries into mathematical formulations and executable code recognizable to external solvers (Zhang et al., 2024a; Tang et al., 2024). Further, the introduction of multi-agent frameworks enables the coordination among a structured sequence of LLM agents to perform tasks including formulation, programming, and evaluation for a target problem (Xiao et al., 2023; AhmadiTeshnizi et al., 2024). Nonetheless, these methods predominantly rely on the intrinsic knowledge embedded within LLMs, which limits their efficacy in addressing problems beyond the scope of their training data. This paper delves into directly generating programs for solving complex VRPs by integrating LLMs' internal knowledge and external references, without the process of mathematical model formulation.

**NCO methods for VRPs.** Beyond LLMs, quite a few approaches automate end-to-end solutions for VRPs through deep (reinforcement) learning, collectively known as neural combinatorial optimization (NCO) (Kool et al., 2019; Kim et al., 2022; Luo et al., 2023). The predominant NCO methods typically

employ Transformer-like neural architectures to process features (e.g., customer coordinates) in VRP instances by encoders and construct VRP solutions (i.e., tours) by the decoder. While these methods bypass the reliance on manually designed heuristics to some extent, the heavy NCO models are often trained separately on individual and simple VRP variants (Hottung et al., 2021; Wu et al., 2021; Zhou et al., 2023a; Goh et al., 2024) with massive time cost. Moreover, the simplified constraint-handling strategies hamper their applicability to complex VRPs with intricate real-world constraints.

## 2.2 RETRIEVAL-AUGMENTED GENERATION

RAG approaches leverage the input sequence to retrieve relevant documents, which are subsequently utilized as supplementary context while generating the target sequence. As a potent mechanism to inject external knowledge into LLMs, the RAG is widely studied for language tasks, such as question answering (QA) (Lewis et al., 2020; Jiang et al., 2023), dialog generation (Shen et al., 2023), and fact verification (Wang et al., 2023). In addition, there are some efforts applying RAG in code generation, which generally retrieve information from different sources, such as web content (Parvez et al., 2021), fixed repository (Zhang et al., 2023), code documentation (Zhou et al., 2023b), or the combination of multiple resources (Su et al., 2024). Interested readers can refer to (Gao et al., 2023) for a thorough and systematic review. VRP solvers usually have elaborate documentation and example codes contributed by the community, which can serve as external knowledge sources for RAG. However, retrieving irrelevant documents is probably unhelpful and even harmful to performance (Yoran et al., 2024). To address this, we decompose the retrieval for separate constraints and progressively refine the documents, which enhances the performance of RAG in generating more accurate programs.

## 3 PRELIMINARIES

### 3.1 VEHICLE ROUTING PROBLEMS

The objective of typical VRPs is to determine a set of vehicle routes with the least cost. The basic constraints are 1) each customer is visited exactly once by a single vehicle, and 2) all vehicles depart from and return to one or more depots (Braekers et al., 2016). Suppose that there is one depot indexed by 0, the commonly used objective for a VRP with $m$ vehicles and $n$ customers is formulated as

$$J = \min \sum_{k \in M} \sum_{i \in N} \sum_{j \in N} c_{ij} x_{ij}^k \tag{1}$$

where $M = \{1, \ldots, m\}$ and $N = \{0, 1, \ldots, n\}$ represent the set of vehicles and the locations of depot and customers, respectively. $c_{ij}$ is the traversal cost between customer $i$ and $j$, and $x_{ij}^k$ is the binary decision variable, indicating if vehicle $k \in M$ traverses from $i$ to $j$.

A typical set of constraints for VRPs is formulated as follows,

$$\sum_{k \in M} \sum_{j \in N} x_{ij}^k = 1 \quad \forall i \in N, i \neq 0 \tag{2}$$

$$\sum_{j \in N} x_{0j}^k = 1 \quad \forall k \in M \tag{3}$$

$$\sum_{i \in N} x_{i0}^k = 1 \quad \forall k \in M \tag{4}$$

$$\sum_{j \in N} x_{ij}^k = \sum_{j \in N} x_{ji}^k \quad \forall i \in N, k \in M \tag{5}$$

where Eq. (2) ensures each customer is visited exactly once by only one vehicle; Eq. (3) and Eq. (4) means vehicles depart from and return to the depot; Eq. (5) ensures the vehicle flow conservation. Besides the above basic constraints, different VRP variants are characterized by various constraints that reflect practical restrictions for vehicle routing in real life.

In this paper, we consider the following additional VRP constraints: 1) *Vehicle capacity*, limiting the maximum load a vehicle can carry; 2) *Distance (or duration) limit*, restricting the total distance or time a vehicle can travel; 3) *Time windows*, requiring vehicles to visit customers within specified

time intervals; 4) *Multiple depots*, allowing vehicles to start and end routes at different depots; 5) *Open route*, where the start and end node of vehicles are not specified; 6) *Prize collecting*, optimizing routes by balancing the penalty of locations that are not visited; 7) *Pickups and deliveries*, managing paired pickup and drop-off demands within a route; 8) *Service time*, accounting for the time spent in serving customers at each location; 9) *Resource constraints*, limiting the number of vehicles that can be loaded or unloaded at the depot simultaneously, potentially causing delays in departure or return. VRP variants featured by combinations of the above constraints are elaborated in Appendix B.

Typically, a VRP, including its objective and constraints, is expected to be formulated as a mathematical program by a human expert. Once the problem is accurately modeled, existing solvers, such as Gurobi (Gurobi, 2024) and OR-Tools (Furnon & Perron, 2024), can be called to solve the given VRP.

## 3.2 Problem Formulation

We solve a code generation (or code completion) problem, without the mathematical model formulation process as done in (Ramamonjison et al., 2022; Xiao et al., 2023; AhmadiTeshnizi et al., 2024). In our approach, the input to an LLM consists of the name of a VRP variant and the corresponding function signature, which specifies the function's name, its parameters, and parameter types. With each parameter in the function described by the docstring, the LLM is responsible for completing the "solve" function by invoking a designated solver. We illustrate an example of the function signature in Appendix B. Compared to using textual descriptions of problems as input (Huang et al., 2024), our formulation offers better generalization for two reasons: 1) once a function is successfully generated, it can be applied to all instances of that specific VRP variant, and 2) only describing basic docstrings reduces the volume of input to an LLM and minimizes the inference effort required for prompting.

Formally, given an input $q$ representing a VRP, an LLM $P(y \mid q)$ generates a program $y$ recognizable to a solver, which can be applied to solve the VRP. We assume the availability of a collection of documents $\mathcal{D}$, where each document corresponds to a part of documentation or example codes for the solver. During the RAG process, the generation is conditioned on a particular subset of documents $\mathcal{D}_s \subseteq \mathcal{D}$. The marginalized generation probability over all $\mathcal{D}_s \subseteq \mathcal{D}$ is given by,

$$P(y \mid q, \mathcal{D}) = \sum_{\mathcal{D}_s \subseteq \mathcal{D}} P(y \mid q, \mathcal{D}_s) \cdot P(\mathcal{D}_s \mid q, \mathcal{D}) \tag{6}$$

As enumerating all possible subsets is computationally infeasible, we use a retriever $\mathcal{R}$ to select the most probable subset of documents $\hat{\mathcal{D}}_s := \arg\max_{\mathcal{D}_s \subseteq \mathcal{D}} P_{\mathcal{R}}(\mathcal{D}_s \mid q, \mathcal{D})$, and thereby enables the LLM to produce a program based on the most likely relevant documents:

$$P(y \mid q, \mathcal{D}) \approx P(y \mid q, \hat{\mathcal{D}}_s) \cdot P(\hat{\mathcal{D}}_s \mid q, \mathcal{D}) \tag{7}$$

## 4 Methodology

Our approach aims to enable LLMs to invoke solvers more accurately for solving VRPs by decomposing the problems and integrating external knowledge. Solving VRPs using LLMs is characterized by the following aspects: 1) Once the generated program is successfully verified on a single instance, it can be applied to all problem instances of the same structure (e.g., the same types of constraints and input parameters). This allows for convenient self-debugging on a simple instance using the LLM and the code executor; 2) The structure of code for addressing different VRPs is mostly the same when calling the same solvers, and the primary variation lies in how constraints are programmed through the solver API functions. These characteristics of LLMs motivate us to perform decomposed retrievals for specific constraints and enhance the quality of code generation. Therefore, we propose the DRoC framework that elegantly amalgamates the two aforementioned points. The framework is illustrated on the left subfigure of Figure 2, which is carried out in the following steps:

- **Step 1: Direct code generation**: An LLM as the first-time generator is prompted directly by the input $q$ (i.e., a VRP) to generate a program $y$, without external information retrieval. Here the code generation purely depends on the internal knowledge of LLM, prompting it to solve the problem by its inherent programming capability.
- **Step 2: Code check**: The program generated in Step 1 is run by a code executor, invoking a solver to solve the VRP. The LLM will be provided with execution traceback if the code contains errors, meaning an injection of external knowledge into the LLM.

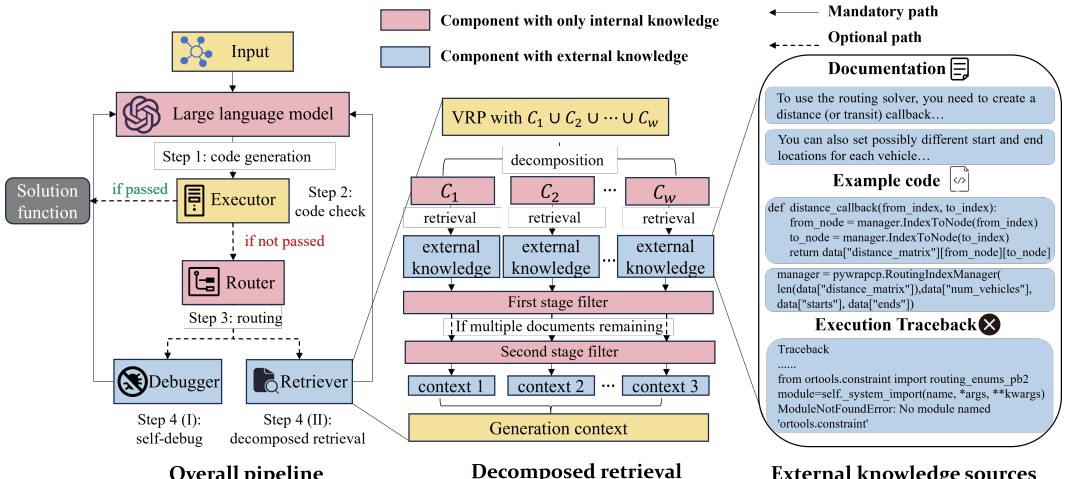

Figure 2: Overview of the proposed DRoC framework.

- **Step 3: Routing**: According to the execution traceback, an LLM as a router determines the operation in **Step 4**, either self-debugging (I) or RAG (II).

- **Step 4 (I): Self-debugging**: An LLM as the self-debugger analyzes the execution traceback (and errors) and attempts to refine the code, which produces a new version of the program.

- **Step 4 (II): Decomposed retrieval**: The retrieval is decomposed to refer LLM to external documentation or example codes for seeking relevant documents on separate constraints of the target VRP, so as to enhance the accuracy of code generation.

In the following subsections, we present the detailed process of the DRoC framework.

## 4.1 EXTERNAL KNOWLEDGE SOURCES

As VRP solvers generally have elaborate documentation and example codes contributed by the community, we incorporate them into our external knowledge sources for retrieval rather than relying solely on single-source data as is common in literature (Zhang et al., 2023; Zhou et al., 2023b). For example, Google's OR-Tools (Furnon & Perron, 2024) provides a detailed tutorial for solving VRPs in its online documentation[1] and has ample example codes in its open-source repository[2]. The multi-source information can be more synergistic and actively utilized by the LLM during RAG.

In addition, we leverage feedback from the code executor (e.g., a Python interpreter) to empower the LLM to precisely identify errors within the code. Unlike the retrieval of documents from other knowledge sources (e.g., documentation and example codes), which typically requires conducting a semantic similarity search in embedding space, obtaining execution feedback involves direct access to information generated by the interpreter (Su et al., 2024), such as error information and traceback.

Using OR-Tools as an example, the external knowledge is briefly shown on the right subfigure of Figure 2. We also investigate a dynamic knowledge source update via Bootstrap for potential performance improvement, which is discussed in Appendix G. Note that external knowledge contains both relevant and irrelevant data, so it is critical to design an effective and precise retrieval mechanism.

## 4.2 DECOMPOSED RETRIEVAL

Despite the availability of documentation and example codes for a solver, such as those for OR-tools, generating accurate programs for a VRP (with composite constraints) is still challenging, due to the difficulty in obtaining an appropriate context to guide LLMs via RAG. On one hand, external

---

[1]https://developers.google.com/optimization/routing
[2]https://github.com/google/or-tools

knowledge sources typically contain exemplar problems with simple constraint structures and may not directly provide documents relevant to the target VRP. On the other hand, the retrieval process may overlook critical constraints if the problem is not properly decomposed. For example, using keywords like "open capacitated vehicle routing problem" often results in retrieving documents related to CVRP, neglecting the key constraint of the open route. This underscores the need for a more nuanced approach to ensure that all relevant constraints are consistently considered. To overcome the issue, we propose to progressively cope with the constraint in a decomposed manner. We break down the retrieval into three sub-processes, including problem decomposition, single-constraint resolution, and context merging. Specifically, we first decompose a target VRP into individual constraints and then resolve these constraints by retrieving from external knowledge sources. Finally, the retrieved documents are merged to form the context for the LLMs, which are used to guide the code generation.

**Problem decomposition.** To formulate queries for retrieval and handle constraints separately, we decompose the target VRP based on its constraints. In addition to the general constraints formulated by Eq. (2)~(5), the VRP variants have their own specific constraints, e.g., the additional constraints described in Section 3.1. Since these constraints are known (Elshaer & Awad, 2020), LLMs have a basic understanding of their meaning. Therefore, we employ a decomposer (i.e., an LLM) to split the constraints of the target VRP into individual items, with each represented by a keyword of the corresponding constraint. As shown on the middle subfigure of Figure 2, $C_1$, $C_2$, ..., $C_w$ are keywords of individual constraints. A VRP with $w$ additional constraints produces $w$ keywords.

**Single-constraint resolution.** The limited internal knowledge of LLMs hinders their ability to accurately generate codes for specific constraints. We enhance them by retrieving relevant external knowledge (i.e., documentation/example codes). We employ OpenAI's embedding model to transform external knowledge into embeddings for dense retrieval. The retriever uses the input "Python code of $C_i$", $i \in \{1, \ldots, w\}$ as query $Q_i$ to conduct a semantic similarity search among all the embedded documents. With the embedding $\mathcal{E}_d$ of each document $d \in \mathcal{D}$ and the embedding $\mathcal{E}_{Q_i}$ of the $i$-th query text, we use squared Euclidean distance to measure the similarity between $Q_i$ and each document $d$:

$$\text{Distance}(Q_i, d) = \sum_{j=1}^{E} (\mathcal{E}_{Q_i}^j - \mathcal{E}_d^j)^2 \tag{8}$$

where $E$ denotes the dimension of the embedding space. The top-$k$ nearest documents are selected by the retriever as the candidates for the corresponding constraint.

Given that external knowledge may contain irrelevant information, we implement a two-stage filter to refine the candidate documents for each constraint. The first stage involves invoking an LLM (i.e., first-stage filter) to assess the relevance between the documents and the constraint $C_i$. By doing so, the LLM is tasked with explicitly articulating the rationale behind the identified documents as relevant, which refer to pertinent documents as supporting evidence. The output is structured into three fields: *relevant*, *code snippet*, and *summary*, with an example in Appendix A.2. The second stage is activated if multiple documents remain after the initial filtering. An LLM (i.e., second-stage filter) is instructed to aggregate the documents and their corresponding summaries, ultimately selecting the most relevant one $\mathcal{D}_i$ for $C_i$ through a comparative analysis fulfilled by the LLM itself.

**Context merging.** After obtaining all the single-constraint contexts, i.e., the most relevant document for each constraint, we simply concatenate them as the merged generation context, which is defined by $\hat{\mathcal{D}}_s = \{\mathcal{D}_1, \ldots, \mathcal{D}_w\}$. The context as part of the input to the LLM is used to generate new programs.

### 4.3 IMPLEMENTATION DETAILS

Given the pipeline of DRoC illustrated in Figure 2, we allow the LLM to generate code up to $I$ iterations, meaning the process will terminate even if a successful program, which outputs feasible solutions to the given VRP, is not obtained after $I$ attempts. Specifically, if the first-time generator fails to produce an appropriate program using only its internal knowledge, a router (i.e., an LLM) is invoked to dynamically choose between two strategies for utilizing external knowledge: self-debugging or decomposed retrieval. We employ two distinct prompt templates to guide the LLM's role in leveraging the retrieved external knowledge: the retrieval-augmented generator and the retrieval-augmented debugger. More precisely, the retrieval-augmented generator is triggered only once, in order to generate a completely new program based on the retrieved context, while the retrieval-augmented debugger is invoked for the remaining $I - 2$ iterations to progressively refine

Table 1: Performance of different methods with gpt-3.5-turbo and gpt-4o. The reported values are averaged over the results of 48 VRP variants.

| Method (gpt-3.5-turbo) | AR ↑ | RER ↓ | Method (gpt-4o) | AR ↑ | RER ↓ |
|---|---|---|---|---|---|
| Standard Prompting | 14.58% | 66.67% | Standard Prompting | 20.83% | 58.33% |
| CoT | 8.33% | 79.17% | CoT | 22.92% | 50.00% |
| PHP | 12.50% | 79.17% | PHP | 22.92% | 54.17% |
| Self-debug | 14.58% | 66.67% | Self-debug | 31.25% | 37.50% |
| Vanilla RAG | 10.42% | 60.42% | Vanilla RAG | 37.50% | 31.25% |
| Self-RAG | 16.67% | 56.25% | Self-RAG | 33.33% | 39.58% |
| DRoC (Ours) | **20.83%** | **47.92%** | DRoC (Ours) | **45.83%** | **20.83%** |

the previously generated code by incorporating insights from external documents. In addition to the RAG processes, the self-debugging operation can also be introduced if the LLM thinks the error can be fixed by itself. This dynamic routing process ensures a more flexible and adaptive framework, improving the likelihood of generating accurate solutions for complex problems.

The prompts for all components in our framework are provided in Appendix A.1, including the first-time generator, router, self-debugger, decomposer, filters, retrieval-augmented generator, and retrieval-augmented debugger. These prompts detail the instructions given to the LLM in the pipeline.

## 5 EXPERIMENTS

To verify the effectiveness of DRoC, we conduct extensive experiments. We evaluate the DRoC and other baselines on 48 VRPs variants by combining different constraints. These VRP variants and the corresponding dataset are elaborated in Appendix B. In principle, the DRoC framework can work with any LLMs or optimization solvers. In our experiments, we mainly use ChatGPT (gpt-4o-2024-08-06 and gpt-3.5-turbo-0125) as the chosen LLM and OR-tools as the optimization solver. In addition, we provide experimental studies on other proprietary and open-source LLMs (i.e., claude3.5 and llama3.1), and another widely used solver (i.e., Gurobi), to show the generalizability. We set the number of retrieved documents $k = 3$ and the number of attempts $I = 4$. The same values for $k$ and $I$ are used across all baselines to ensure a fair comparison. The best result among 3 independent runs is reported for all the methods. We use the following two performance metrics:

- Accuracy Rate (AR): This metric is defined as AR $= \frac{V_a}{V_t} \times 100\%$, where $V_t$ is the total number of generated programs for different VRP variants, and $V_a$ represents the number of successful programs that result in the optimal solution for a given VRP variant.

- Runtime Error Rate (RER): This metric is defined as RER $= \frac{V_e}{V_t} \times 100\%$, where $V_e$ indicates the number of program that encounter runtime errors, representing the proportion of generated programs experiencing execution errors stemming from internal logical errors because of incorrect calls of solver APIs or syntax errors.

### 5.1 BASELINES

We benchmark DRoC against 6 baselines in the main results: Standard Prompting, Chain-of-Thought (Wei et al., 2022), Progressive-Hint Prompting (PHP) (Zheng et al., 2023), Self-debug (Chen et al., 2024), Vanilla RAG (VRAG), and Self-RAG (Asai et al., 2024). In addition, we compare DRoC with two recent works, Evolution of Heuristics (EoH) (Liu et al., 2024) and Reflective Evolution (ReEvo) (Ye et al., 2024), which use LLMs to improve heuristics via evolutionary computation. We name them LLM+EC methods. More details of the baselines are elaborated in Appendix C. Our code and data are publicly available at https://github.com/Summer142857/DRoC.

### 5.2 OVERALL PERFORMANCE

Table 1 presents the performance of the proposed DRoC and 6 baselines in terms of AR and RER. The results show that although applying a more powerful LLM (i.e., gpt-4o) does improve the performance of all tested methods, all 6 baselines were able to produce successful programs only for less than 40%

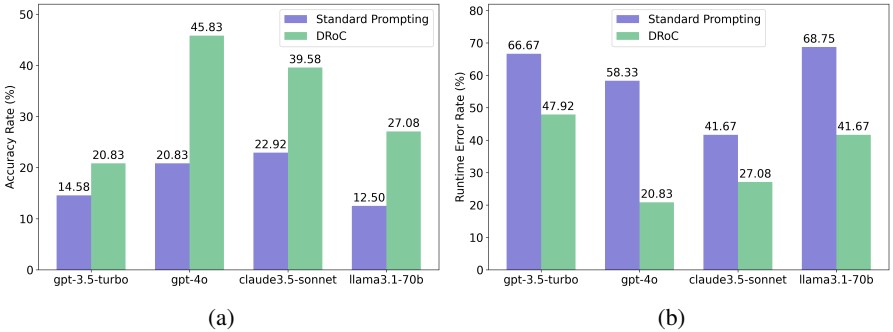

Figure 3: Performance of DRoC and Standard Prompting with different LLMs: (a) AR metric (b) RER metric. The DRoC is applicable to varied LLMs, showing clear performance enhancements.

of tested VRP variants. This demonstrates the difficulty in solving complex NP-hard problems for SOTA LLMs. We observe that the methods that either rely solely on the internal knowledge of LLMs (i.e., Standard Prompting, CoT, and PHP) or only combine execution feedback (i.e., Self-debug) do not result in good performance. Meanwhile, the performance boost from VRAG and Self-RAG is minimal, and VRAG can actually lead to performance degradation (for gpt-3.5-turbo), suggesting that inappropriate or ineffective retrieval methods fail to provide significant assistance in solving VRPs.

In comparison, the proposed approach achieves the best results in both generating correct programs and reducing runtime errors. Compared to the standard prompting approach, DRoC successfully solves 25% more VRP variants by gpt-4o. Moreover, it produces more generation of error-free programs. For solutions that do not have running errors but are not optimal, they may miss some of the constraints, and human coder can further improve them without building from scratch. More illustrative results of the generated solutions are provided in Appendix D, where we present visual plots of the solutions for various VRP instances. Additionally, we also compare the incorrect and correct API-calling code generated before and after applying our method. DRoC also demonstrates applicability to other OR problems, such as the assignment and facility location problems, which we discuss in detail in Appendix E. This highlights the need for more refined retrieval techniques and integration strategies, as in DRoC, for tackling complex problem-solving scenarios.

Meanwhile, we report the running time of the generated programs in Appendix F and find that DRoC does not lead to any significant increase in running time of the problem-solving process.

## 5.3    EVALUATION WITH DIFFERENT LLMS

To demonstrate that the DRoC is a general tool for enhancing VRP-solving capabilities with LLMs, we also evaluate its performance with the other two LLMs: claude-3.5-sonnet-20241022 and llama3.1-70b. The results are presented in Figure 3. We observe that even advanced LLMs, such as gpt-4o and claude-3.5-sonnet, still struggle to correctly solve VRPs. However, the proposed DRoC consistently improves the performance of various LLMs, indicating that DRoC can function as a generic tool to enhance the VRP-solving abilities of LLMs in spite of their different architectures.

## 5.4    EVALUATION WITH GUROBI SOLVER

We show DRoC can embed different optimization solvers such as the popular Gurobi solver. Different from OR-tools, which solves VRPs by simply calling the APIs, the use of Gurobi for solving a particular VRP variant requires us to first build the corresponding Mixed-Integer Programming (MIP) model, making it a more difficult task. In the experiments, we use the programs of 10 VRP variants, which only contain 0 or 1 additional constraint, as the external knowledge source, and allow the DRoC to retrieve from these simple VRP solutions. We evaluate the performance on advanced LLMs, i.e., gpt-4o and claude-3.5-sonnet. The results (see Table 2) show that DRoC remains effective when working with the Gurobi solver. While we only use VRPs with a single constraint as external knowledge, the LLMs can accurately solve more VRP variants with more than one constraint, indicating that complex tasks can be fulfilled by our decomposition-based method.

Table 2: The performance evaluated on Gurobi solver with and without DRoC.

| LLM | AR ↑ | RER ↓ |
|---|---|---|
| gpt-4o (Standard Prompting) | 2.08% | 45.83% |
| claude-3.5-sonnet (Standard Prompting) | 12.50% | 52.08% |
| gpt-4o (DRoC) | **31.25%** | **25.00%** |
| claude-3.5-sonnet (DRoC) | **31.25%** | **37.50%** |

## 5.5 ABLATION STUDY

Table 3: The results of ablation studies.

| Method | OR-tools | | Gurobi | |
|---|---|---|---|---|
| | AR ↑ | RER ↓ | AR ↑ | RER ↓ |
| DRoC (Full) | **45.83%** | **20.83%** | **31.25%** | **25.00%** |
| w/o filter | 37.50% | 31.25% | 27.08% | 27.08% |
| w/o DR | 35.42% | 29.17% | 27.08% | 29.17% |
| w/o router | 43.75% | 29.17% | 29.17% | 31.25% |

We conduct ablation studies for both OR-tools and Gurobi for a more comprehensive comparison. The studies are based on gpt-4o, which has showcased good performance under different settings.

**Ablation study on two-stage filter.** We first evaluate the necessity of the filter process, which refines the retrieved documents and reduces extraneous information. As shown in Table 3, we observe a slight drop in model performance when potentially irrelevant documents are not filtered out. This outcome is similar to the poorer performance observed with VRAG shown in Table 1, suggesting that the relevance of the context provided during generation significantly impacts the results. The filter ensures that pertinent information is used, which is crucial for optimizing VRP-solving effectiveness.

**Ablation study on decomposed retrieval (DR).** In order to evaluate the necessity of DR, we replace it by direct retrieval of documents, which takes "Python code of {the name of the VRP}" as the query, aiming at retrieving code that is mostly close to the target VRP variant. This replacement is applied whenever the retriever is called, and the final context is obtained by randomly choosing from top-$k$ retrieved documents. Similarly, there is also a performance drop for both OR-tools and Gurobi, suggesting that LLM can learn to solve complex VRPs from single-constraint resolutions in the DR.

**Ablation study on router.** We replace the router with a random routing strategy, which randomly routes the workflow to the self-debugger or retrieval-augmented debugger. There is also a slight drop in performance without the proposed router, indicating that the selection between execution-based and documentation-based external knowledge is important.

## 5.6 COMPARISON WITH LLM+EC METHODS

LLMs can be used to evolve heuristics for solving VRPs, as shown in the literature. We conducted experiments to find out how such an approach performs in comparison to our approach which is based on VRP solvers. We take the Prize Collecting Travelling Salesman Problem (PCTSP) as a demonstration problem, which ChatGPT cannot originally solve due to the incorrect calls of solver API (see Appendix D), to conduct a comparison study between SOTA LLM+EC methods and the proposed DRoC.

We utilize EoH and ReEvo to evolve the ant colony algorithm, as detailed in (Ye et al., 2024), and compare the results of these evolutionary approaches. Specifically, we record both the best objective values and the number of tokens consumed by the LLM for EoH

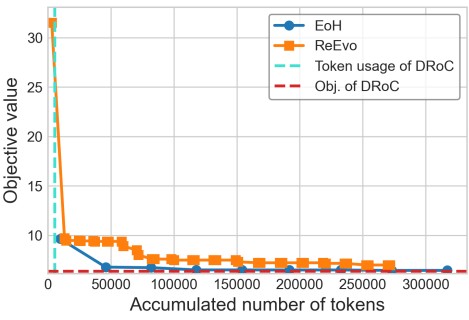

Figure 4: Comparison between LLM+EC methods (EoH and ReEvo) and DRoC.

and ReEvo during iteration-based evolution. As shown in Figure 4, compared to DRoC, the LLM+EC methods require a substantial number of tokens (e.g., over 0.1M) to evolve towards a solution which significantly increases computational costs and potential carbon emissions. Notably, the best heuristics for EoH and ReEvo achieve objective values of 6.436 and 6.984, respectively, while DRoC with OR-tools yield a superior result of 6.352. The findings suggest that our DRoC framework is more efficient and competitive than EC methods, providing greater enhancement of the LLM.

## 5.7 SENSITIVITY ANALYSIS

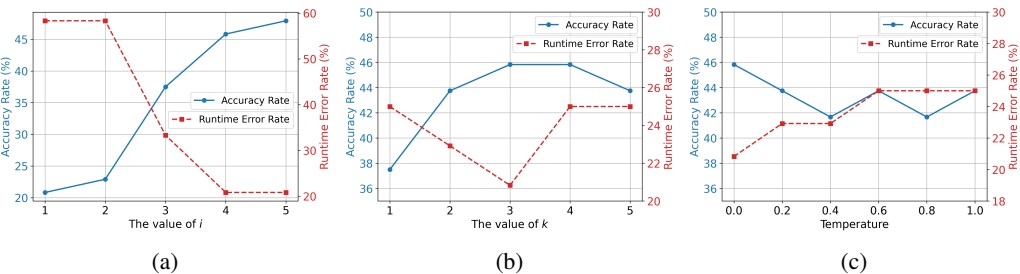

Figure 5: The results for sensitivity analysis on (a) $I$; (b) $k$; (c) temperature.

We study how three key parameters influence the performance of DRoC: the maximum number of generation $I$, the number of retrieved documents $k$, and the temperature of the LLM. The analysis is also based on gpt-4o, and the results are shown in Figure 5.

**Sensitivity analysis on $I$.** The performance of DRoC generally improves with the increase of $I$, but the improvement turns marginal from 4 to 5. Therefore we set $I = 4$ across all our main experiments.

**Sensitivity analysis on $k$.** The value of $k$ has less influence than $I$. The performance is slightly improved when varying $k$ from 1 to 3, mainly because more comprehensive contents are retrieved with a larger $k$. After that, the performance tends to be stable because the generation context is relatively unchanged since redundant documents are filtered out by the two-stage filter process.

**Sensitivity analysis on temperature.** The performance of DRoC remains relatively stable across different temperature parameters. This indicates that the combination of iterative refinement and targeted document selection helps maintain consistent results, regardless of variations in the randomness of generation influenced by the temperature configuration.

## 5.8 BOOTSTRAP-BASED OPTIMIZATION

As the LLMs can solve more problems utilizing external knowledge, they can also take the correct generation as part of the external knowledge, making it possible to improve the performance through Bootstrap. We also analyze the impact of such a Bootstrap mechanism and find that the integration of LLM generations and original external knowledge (publicly accessible documentation and codes) can also boost the accuracy to some extent. The details and result are elaborated in Appendix G, and we find that more than 50% VRP variants can be resolved after introducing the Bootstrap mechanism.

## 6 CONCLUSIONS

In this paper, we propose DRoC, an effective framework designed for solving VRPs with complex constraints, utilizing LLMs and optimization solvers. By integrating external knowledge through retrieval-augmented generation and decomposing constraints for more accurate retrieval, the DRoC significantly improves LLM performance across a wide range of VRP variants. For instance, it improves the accuracy rate of gpt-4o from 20.83% to 45.83%. In the future, we plan to expand our focus to integrate the DRoC with solver selection and metaheuristic generation processes, enhancing the versatility and reliability in addressing more diverse optimization tasks. We will also introduce more external knowledge sources for better RAG performance and integrate modeling function into our framework, further boosting its effectiveness.

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

# A PROMPT AND OUTPUT TEMPLATES

## A.1 PROMPTS

Table 4: The prompts skeleton for each DRoC component

| Component | Prompt Skeleton |
|---|---|
| First-time generator | You are an expert in Python programming for operations research. You are very good at calling solver in Python and solving problems. Respond with the syntactically correct code for solving a problem using solver. Make sure you follow these rules: 1. Read the template. First understand the meaning of the parameters in 'solve' function, and then complete the code inside the function. 2. Ensure all parameters in the template are used in the function. 3. Do not give additional examples or define main function for testing. 4. Return the objective value of the problem by the 'solve' function. 5. Ensure any code you provide can be executed with all required imports and variables defined. Template: {code_example} Structure your answer with a description of the code solution, and then list the imports, and finally list the functioning code block. |
| Router | Your task is to determine how to refine the incorrect Python code, which is produced by another programmer. Here is the code: <prep_code> The code is about solving a problem based on solver, and there is the error information while running the code: Error message: <message> There are several tools that can be called, which can be one of the following: (1) retrieval_augmented_debug[input]: Retrieve code examples from a repository, and then refine the current program drawing upon the retrieved codes. Prioritize it when the error is caused by incorrect use of solver API. (2) self_debug[input]: Call a pretrained LLM like yourself. Prioritize it when you are confident in fixing the error yourself, e.g., when the error of the code is caused by syntax error or wrong import. Return "1" if you think you should use tool (1), otherwise return "2". Do not return other things. |
| Self-debugger | You are an expert in Python programming for operations research by calling solver. Now your responsibility is to debug the code snippet with errors. The code snippet with bug is as <prep_code>. Here is the error message of the code: <message>. You can first reason about the error, and finally refine the code and return the whole fixed function. Ensure any code you provide can be executed with all required imports and variables defined. Remember, the final solution should be returned by the 'solve' function. Do not use other name for the function and do not give example usage of the function. Structure the refined solution by firstly giving the reason of the error and the strategy for fixing it. Then list the imports. Finally list the functioning code block and solve the problem with 'solve' function. |
| Decomposer | You will extract the keywords of a vehicle routing problem (VRP) for me. I give you the name of a VRP and you produce the keywords according to its constraints. Structure your answer with a list of keywords inside "<>" and use commas to separate different keywords. Do not return other things. For example, the output of "Capacitated Vehicle Routing Problem with Time Windows and Multiple Depots (CVRPTWMD)" should be <Capacitated, Time Windows, Multiple Depots>, and the output of "Prize Collecting Travelling Salesman Problem (PCTSP)" should be <Prize Collecting>. |

| First-stage filter | You are an expert in Python programming and solver for vehicle routing problem.
I will give you a retrieved documents potentially related to keyword, and you will firstly assess if the document includes Python code to program keyword. If so, you should explain how the code addresses the constraint of keyword. Here is the retrieved document:
{context}
If the document contains Python code related to keyword, grade it as relevant. After that, extract the code snippet in the document related to keyword. Finally, produce an explanation on how to program the constraint of keyword, and your goal is to make other programmers know how to do that. Structure your answer with the binary score 'yes' or 'no' to indicate whether the document is relevant, and then list the related code snippet, and finally give the summary.
If the document is not related, just return 'no' for the binary score, and nothing for the code snippet and the summary. |
|---|---|
| Second-stage filter | You are an expert in Python programming and solver for vehicle routing problem (VRP).
I will give you several retrieved documents (codes) and their explanations potentially related to keyword, and you should assess which context is the most relevant one and with minimal redundant information.
Here are the documents, which are separated by '====================':
{contexts}
Return the index of the most relevant document and do not return anything else. For example, if you think the second document is the most relevant one, just return 2. Please strictly return integer index following the above instruction. |
| Retrieval-augmented generator | You are an expert in Python programming for operations research and combinatorial optimization. You are good at calling <solver> in Python and solving problems.
Respond with the syntactically correct code for solving a problem using solver. Make sure you follow these rules: 1. Read the template. First understand the meaning of the parameters in 'solve' function, and then complete the code inside the function.
2. The context provides example codes of addressing each constraint of {problem} by {solver}. Learn to model each constraint and solve the problem accordingly.
3. Do not give additional examples or define main function for testing. 4. Return the objective value of the problem by the 'solve' function. 5. Ensure any code you provide can be executed with all required imports and variables defined.
Template: {code_example}
Context: {context}
Structure your answer with a description of the code solution, and then list the imports, and finally list the functioning code block. |
| Retrieval-augmented dubugger | You are responsible for refining the code with errors, which tries to solve problem by calling solver in Python.
The code snippet with the bug is as <prep_code>.
Here is the error message of the code: <message>.
Make sure you follow these rules: 1. You can first reason about the error, and then refine the code and return the whole fixed function.
2. The context provides examples of solving problems with different constraints, referring to the relevant parts and modifying the code accordingly: <context>.
3. Do not give additional examples or define the main function for testing.
4. Return the objective value of the problem by the 'solve' function.
5. Ensure any code you provide can be executed with all required imports and variables defined.
Structure your answer with a description of the code solution, then list the imports, and finally list the functioning code block. |

## A.2 OUTPUT EXAMPLE OF THE FILTER

```
relevance="yes",
code_snippet="# Add Capacity constraint\n
def demand_callback(from_index):\n
# Convert from routing variable Index to demands NodeIndex\n
from_node = manager.IndexToNode(from_index)\n
return demands[from_node]\n\n
# Register the demand callback with the routing model\n
demand_callback_index = routing.RegisterUnaryTransitCallback(
demand_callback)\n    routing.AddDimensionWithVehicleCapacity(
demand_callback_index, 0, # null capacity slack\n
vehicle_capacities,  # vehicle maximum capacities\n   True,  # start
cumul to zero\n        'Capacity')",
 summary="To program the Capacitated constraint in the Capacitated
Vehicle Routing Problem with Distance Limit (CVRPL) using OR-tools in
 Python, you need to define a demand callback function that maps the
routing variable Index to demands NodeIndex. This function is
registered as a unary transit callback with the routing model. Then,
the capacity constraint is added using the
AddDimensionWithVehicleCapacity method, specifying the demand
callback index, null capacity slack, vehicle maximum capacities,
start cumul to zero, and the dimension name 'Capacity'. This ensures
that the vehicle capacities are respected during the routing
optimization process."
```

Figure 6: The example of the output of the first-stage filter.

## B  VRP VARIANTS

```python
def solve(time_matrix: list, time_windows: list, demands: list,
          vehicle_capacities: list, num_vehicles: int,
          starts: list, ends: list):
    """
    Args:
        time_matrix: contains the integer travel times between locations
        time_windows: the list of tuples for time windows of the
    customers
        demands: the list of integer customer demands
        vehicle_capacities: the capacity of each vehicle
        num_vehicles: the number of the vehicle
        starts: the index of the starting depots for vehicles
        ends: the index of the ending depots for vehicles

    Returns:
        obj: a number representing the objective value of the solution
    """
    obj = -1
    return obj
```

Figure 7: Function template of CVRPTWMD.

As mentioned in Section 3.1, there are several common additional constraints for the VRP, and we create VRP variants by combining different constraints, for which is shown in Table 5. Due to the high complexity of VRPs, designing numerous large instances with feasible solutions is challenging. Therefore, we use a carefully crafted, simple instance for each VRP variant, developed by human experts. This approach ensures the feasibility of each instance and provides the dataset for evaluating the performance of various baselines alongside our proposed method, DRoC. The ground truth for the optimal solutions of the instances is primarily obtained using Hybrid Genetic Search (HGS) (Wouda

et al., 2024), which is widely recognized as a method for calculating approximate optimal solutions for VRPs. For VRP variants with constraints of "Pickup and Delivery" or "Resource Constraints," we utilize OR-Tools with a search time limit of 100 seconds to determine their ground truth, as the used HGS solver does not support these specific constraints. To make the instances more informative, we randomly use a distance matrix or a time matrix to represent the graph of the VRP. Therefore, we impose distance limits on those with distance matrix and duration limit on those with time matrix. Accordingly, we create 48 unique VRP variants, each paired with specific instances, forming a benchmark dataset for evaluation. A program is deemed correct when its generated solve function yields the same value as the optimal solution. Additionally, such correct programs can generalize to any VRP instance sharing the same constraint structure, regardless of input parameters (e.g., customer coordinates).

Different from previous studies (Zhang et al., 2024; Huang et al., 2024), which try to solve OR problems with natural language description, we just take the name of the problem and the function signature as input. We take the function signature of the CVRPTWMD as an example, which is shown in Figure 7.

In this case, the LLM needs to try to understand the meaning of each parameter and generate programs accordingly. Once a program for a VRP variant is produced successfully, it can be used in all instances of the same VRP. Compared to natural language-based description, which specifies the data of the problem, this method is more generalizable.

## C BASELINES

In this section, we elaborate on the implementations of the baselines involved in the experiments:

**Standard Prompting**: it refers to using the prompt skeleton of the first-time generator in Section A.1. The generator is called up to $I$ times independently without the injection of any external knowledge.

**Chain-of-Thought**: similar to the CoT baseline in (Xiao et al., 2023), we add the sentence "Let's think step by step" in the standard prompting to guide the model's thought process, aiming at using the internal knowledge of the LLMs for reasoning as much as possible.

**Progressive-Hint Prompting**: similar to the PHP baseline in (Xiao et al., 2023), we produce an initial program and then use previous generations as hints to progressively guide the LLM toward the correct solutions. It is fulfilled by verifying if the current response is the same as the previous one.

**Self-debug**: it is based on the method proposed by Chen et al. (2024), using the error information and corresponding traceback produced by the executor to teach the LLM conduct debug without any human feedback on the code correctness. Specifically, it follows the prompt of the self-debugger in Section A.1. The number of generations is also up to $I$.

**Vanilla RAG**: The VRAG approach retrieves relevant context before each round of program generation. In the first iteration, the query is set as "Python code of the name of the VRP." For subsequent iterations, the query consists of the generated code from the previous iteration to retrieve the most relevant documents. During program generation, the top-$k$ retrieved documents are included as part of the input to guide the model in generating a more accurate solution.

**Self-RAG**: Originally proposed by Asai et al. (2024), we adapt Self-RAG to the VRP tasks. Similar to VRAG, a retriever is used to obtain relevant documents, followed by a relevance grader to assess whether each retrieved document is pertinent to the target VRP. We implement this process using the first-stage filtering mechanism from our DRoC framework. The remaining relevant documents are then used in parallel to generate solutions. Each generated program is executed until one can run successfully. Additionally, the code generated in previous iterations is used as a query for further retrieval, continuing until the maximum number of generations $I$ is reached.

**EoH**: EoH evolves the codes of heuristics by diverse prompt strategies. We basically follow the configuration in the original paper (Liu et al., 2024). We use 30 populations at the initial stage and 10 populations for each iteration. We allow for at most 300 times of evaluations on the PCTSP instances.

**ReEvo**: ReEvo uses the reflection mechanism to progressively evolve the heuristics. We follow the default settings in Ye et al. (2024) with also up to 300 evaluations on the instances.

Table 5: The studied 48 VRP variants with nine additional constraints.

| | Vehicle Capacity | Distance Limit | Time Window | Multiple Depots | Open Route | Prize Collecting | Pickup and Delivery | Service Time | Resource Constraint |
|---|---|---|---|---|---|---|---|---|---|
| TSP | | | | | | | | | |
| TSPTW | | | ✓ | | | | | | |
| TSPTWS | | | ✓ | | | | | ✓ | |
| VRP | | | | | | | | | |
| VRPL | | ✓ | | | | | | | |
| VRPMD | | | | ✓ | | | | | |
| VRPS | | | | | | | | ✓ | |
| VRPSL | | ✓ | | | | | | ✓ | |
| VRPTW | | | ✓ | | | | | | |
| VRPTWL | | ✓ | ✓ | | | | | | |
| VRPTWMD | | | ✓ | ✓ | | | | | |
| VRPTWS | | | ✓ | | | | | ✓ | |
| VRPTWMDL | | | ✓ | ✓ | | | | ✓ | |
| VRPTWSL | | ✓ | ✓ | | | | | ✓ | |
| VRPTWMRC | | | ✓ | | | | | | ✓ |
| VRPTWMRCL | | ✓ | ✓ | | | | | | ✓ |
| CVRP | ✓ | | | | | | | | |
| CVRPL | ✓ | ✓ | | | | | | | |
| CVRPTW | ✓ | | ✓ | | | | | | |
| CVRPMD | ✓ | | | ✓ | | | | | |
| CVRPTWL | ✓ | ✓ | ✓ | | | | | | |
| CVRPMDL | ✓ | ✓ | | ✓ | | | | | |
| CVRPTWMD | ✓ | | ✓ | ✓ | | | | | |
| CVRPTWMDL | ✓ | ✓ | ✓ | ✓ | | | | | |
| CVRPTWRC | ✓ | | ✓ | | | | | | ✓ |
| CVRPTWRCL | ✓ | ✓ | ✓ | | | | | | ✓ |
| PCTSP | | | | | | ✓ | | | |
| PCTSPTW | | | ✓ | | | ✓ | | | |
| PCVRP | | | | | | ✓ | | | |
| PCVRPTW | | | ✓ | | | ✓ | | | |
| PCVRPMD | | | | ✓ | | ✓ | | | |
| PCVRPTWMD | | | ✓ | ✓ | | ✓ | | | |
| OVRP | | | | | ✓ | | | | |
| OVRPL | | ✓ | | | ✓ | | | | |
| OVRPTW | | | ✓ | | ✓ | | | | |
| OCVRP | ✓ | | | | ✓ | | | | |
| OCVRPL | ✓ | ✓ | | | ✓ | | | | |
| OCVRPTW | ✓ | | ✓ | | ✓ | | | | |
| PDP | | | | | | | ✓ | | |
| PDPL | | ✓ | | | | | ✓ | | |
| PDPTW | | | ✓ | | | | ✓ | | |
| PDPMD | | | | ✓ | | | ✓ | | |
| PDPTWL | | ✓ | ✓ | | | | ✓ | | |
| PDPTWMD | | | ✓ | ✓ | | | ✓ | | |
| PDPSL | | ✓ | | | | | ✓ | ✓ | |
| PDPTWS | | | ✓ | | | | ✓ | ✓ | |
| PDPTWSL | | ✓ | ✓ | | | | ✓ | ✓ | |
| PDPTWMDL | | ✓ | ✓ | ✓ | | | ✓ | | |

For the comparison study of EoH and ReEvo, The evolution is conducted on 10 PCTSP instances with 50 nodes, which are randomly sampled from a unit square. Let the distance between node $i$ and the depot be $d_i$, and $d_{\max} = \max_i(d_i)$, the prize of node $i$ is set to $\text{prize}_i = \frac{1 + \lfloor 99 \cdot r \rfloor}{4 \max_i(1 + \lfloor 99 \cdot r \rfloor)}$ where $r = \frac{d_i}{d_{\max}}$. The penalty values for unvisited nodes are set the same as the prizes.

## D   GENERATED SOLUTIONS

We present several examples of solutions that our DRoC method can achieve, which the standard approach fails to generate, as illustrated in Figure 8. These VRPs often involve multiple constraints that pose significant challenges for LLMs to address effectively.

We use gpt-4o to invoke OR-tools for solving VRPs with the Prize Collecting constraint. The primary distinction between the Standard Prompting and DRoC methods lies in how they handle the constraint, with the former failing to produce a correct solution, while the latter succeeds. As shown in Figure 9, the programming approaches for the Prize Collecting constraint differ significantly. DRoC enables

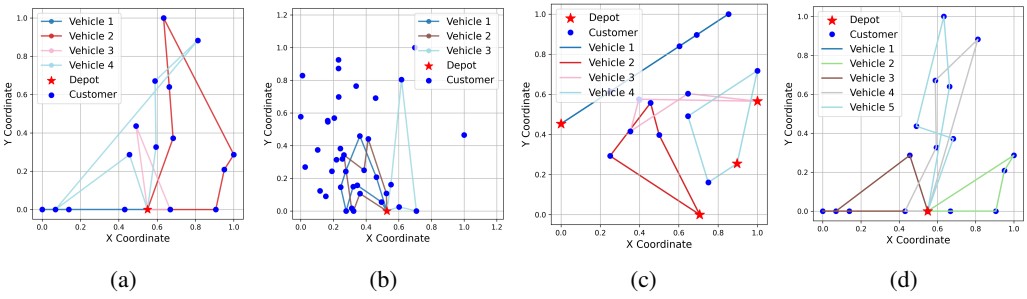

Figure 8: Example solutions generated by DRoC. (a) CVRPTWRC solved by OR-tools; (b) PCVRP solved by OR-tools; (c) CVRPMD solved by Gurobi; (4) VRPTWL solved by Gurobi.

vehicles to drop nodes, effectively accommodating the constraint. In contrast, the standard method produces meaningless content, leading to hallucinations during the generation.

```python
# Define the prize collection callback
def prize_callback(from_index):
    from_node = manager.IndexToNode(from_index)
    return prizes[from_node]
prize_callback_index = routing.RegisterUnaryTransitCallback(
    prize_callback)

routing.AddDimensionWithVehicleCapacity(
    prize_callback_index,
    0,  # no slack
    [sum(prizes)] * num_vehicle,  # vehicle maximum prize capacity
    True,  # start cumul to zero
    'Prize')
# Setting the objective to maximize the prize collection
prize_dimension = routing.GetDimensionOrDie('Prize')
for vehicle_id in range(num_vehicle):
    routing.SetFixedCostOfVehicle(-sum(prizes), vehicle_id)
```

(a) Generated code snippet by Standard Prompting. (Incorrect)

```python
# Allow to drop nodes.
for node in range(1, len(distance_matrix)):
    routing.AddDisjunction([manager.NodeToIndex(node)], prizes[node])
```

(b) Generated code snippet by DRoC. (Correct)

Figure 9: Comparison of code snippets for Prize Collecting constraint.

## E  APPLICATION TO OTHER OR PROBLEM

Many real-world OR problems possess composite constraints, and thus these in which our methods can be applied. In order to show the applicability of the proposed DRoC to other OR problems, we take the assignment problem and facility location problems as the examples, which are both fundamental combinatorial optimization problems.

### E.1  ASSIGNMENT PROBLEM

We focus on the scenario of worker assignment problem: Given assignment costs for each worker-task pair, we need to allocate workers to distinct tasks so that each worker is assigned to one and at most one task and the total cost is minimized. The following additional constraints are considered: 1) *Teams of Workers*: the workers are divided into several teams, and each team can perform at most $x$

tasks, where $x$ is a parameter; 2) *Task sizes*: Each task is assigned a size indicating the necessary time or effort, and the total size of tasks allocated to each worker is restricted by a fixed limit.

```python
# Define constraints
for j in range(num_tasks):
    solver.Add(sum(x[i, j] for i in range(num_workers)) == task_sizes[j])

for i in range(num_workers):
    solver.Add(sum(x[i, j] for j in range(num_tasks)) <= team_max)
    solver.Add(sum(x[i, j] * task_sizes[j] for j in range(num_tasks)) <=
    size_max)
```

(a) Generated code snippet by Standard Prompting. (Incorrect)

```python
# Each worker is assigned to at most one task.
for worker in range(num_workers):
    model.Add(
        sum(task_sizes[task] * x[worker, task] for task in range(
    num_tasks))
        <= size_max
    )

# Each task is assigned to exactly one worker.
for task in range(num_tasks):
    model.AddExactlyOne(x[worker, task] for worker in range(num_workers))

# Each team takes at most team_max tasks.
for team in worker_teams:
    team_tasks = []
    for worker in team:
        for task in range(num_tasks):
            team_tasks.append(x[worker, task])
    model.Add(sum(team_tasks) <= team_max)
```

(b) Generated code snippet by DRoC. (Correct)

Figure 10: Comparison of code snippets for the worker assignment problem.

We consider the worker assignment problem with both constraints of *Teams of Workers* and *Task sizes*. We also use the OR-Tools to solve it. The way that gpt-3.5-turbo handles the constraints is shown as Figure 10 (a), while the code snippet of DRoC is shown in Figure 10 (b). We find that the LLM without DRoC neither groups workers effectively nor assigns the correct number of worker for the tasks. In comparison, our method generates the program by modeling the constraints correctly.

## E.2 FACILITY LOCATION PROBLEM

The Facility Location Problem (FLP) encompasses optimization tasks aimed at selecting optimal sites for $p$ facilities to minimize the conveying cost from facilities to customers. FLPs may include constraints such as facility capacity. Various real-world scenarios present different optimization objectives for FLPs, leading to distinct problem variants, such as: 1) *p-median*: minimizing the total distance from each customer to their nearest facility; 2) *p-center*: minimizing the maximum cost any customer must spend to reach the nearest facility; 3) *p-maxian*: maximizing the total weighted travel cost incurred by all customers; 4) *p-dispersion*: maximizing the smallest distance between any pair of facilities. These objectives necessitate additional constraints during the modeling. Hence, we explore the integration of capacity constraints with these objectives and assess the performance of DRoC on these FLP variants. We take part of the codes implemented by Gaboardi (2015) as the external knowledge and use Gurobi to model and solve the FLP variants.

Table 6 illustrates that DRoC facilitates the resolution of more FLP variants, including some not encompassed by external knowledge. In particular, DRoC augmentation empowers gpt-3.5-turbo to successfully tackle five additional problem variants, while the standard gpt-3.5-turbo exhibits very limited proficiency in solving FLPs.

Table 6: The evaluation results on the FLPs.

| | Contained in the external knowledge | Solved by gpt-3.5-turbo | Solved by gpt-4o | Solved by gpt-3.5-turbo (DRoC) | Solved by gpt-4o (DRoC) |
|---|---|---|---|---|---|
| $p$-median FLP | ✓ | ✗ | ✓ | ✗ | ✓ |
| $p$-center FLP | ✓ | ✗ | ✓ | ✓ | ✓ |
| $p$-maxian FLP | ✓ | ✗ | ✓ | ✓ | ✓ |
| $p$-dispersion FLP | ✓ | ✗ | ✗ | ✓ | ✓ |
| Capacitated FLP | | ✗ | ✓ | ✗ | ✓ |
| Capacitated $p$-median FLP | | ✗ | ✗ | ✓ | ✓ |
| Capacitated $p$-center FLP | | ✗ | ✓ | ✗ | ✓ |
| Capacitated $p$-maxian FLP | ✓ | ✗ | ✗ | ✗ | ✓ |
| Capacitated $p$-dispersion FLP | | ✗ | ✗ | ✓ | ✓ |

## F  RUNNING TIME OF THE GENERATED CODE

Table 7: Comparison of running time

| Method | Mean Time (s) | Std Time (s) | Mean Log Ratio | Std Log Ratio |
|---|---|---|---|---|
| HGS | 1.005 | 0.003 | - | - |
| DRoC (HGS) | 0.329 | 0.713 | -3.257 | 0.317 |
| OR-Tools | 12.503 | 32.379 | - | - |
| DRoC (OR-Tools) | 1.411 | 4.059 | -1.145 | 0.882 |

We also explore the running time of the generated programs and check if the LLM produce code with extended compuation time duration. In our implementations, we used the HGS solver for obtaining optimal solutions and OR-Tools in our RAG framework, respectively. Specifically, the running time for the HGS solver is supposed to be configured, which serves as the stopping condition for the search process. We set a maximum run time of 1 second for HGS, as we found it is sufficient to obtain optimal solutions for the simple instances used in our evaluation. For the OR-Tools solver, we set the search time limit as 100s, as it has an early stop mechanism once there is no solution improvement. The statistics for running time are shown in Table 7. Note that we calculate the running time of DRoC for instances solvable by HGS [denoted as DRoC (HGS)] and those solvable by OR-Tools [denoted as DRoC (OR-Tools)], respectively.

The results empirically show that the mean and standard deviation of the running time for DRoC (OR-Tools) are smaller than those of directly calling OR-Tools. This is mainly because the search limit parameter is not configured for most of the instances in the programs generated by DRoC, while we predetermine the search limit for OR-Tools when calculating the approximated optimal solutions. When we set the same time limit for both OR-Tools and DRoC (OR-Tools), the time metrics can be nearly identical (with a mean log ratio of 0.004). This is intuitive, as the generated code produced by DRoC maintains a similar structure to manually written code for solving specific VRPs. Thus, DRoC does not introduce any significant increase in running time, while it provides an automatic and user-friendly approach to solve the problems.

## G  BOOTSTRAP-BASED OPTIMIZATION

We have introduced DRoC using static external knowledge sources. However, as LLMs, powered by DRoC, begin generating more accurate solutions, we can dynamically update the external knowledge by incorporating these generated solutions. Specifically, we first solve all solvable VRP variants using the static DRoC approach, and subsequently embed all the generated programs, which have been executed successfully, to the knowledge base. We create a new retriever for these LLM-generated solutions and ensemble it with the retriever of other knowledge. Following this, we initiate a new round of generation aimed at solving the previously unsolved problems. By leveraging the solutions generated by the LLMs, we enhance the model's performance in a Bootstrap-based manner, a process we term DRoC with Bootstrap-based optimization (DRoC-BBO). This iterative approach allows the LLM to improve progressively by utilizing its own outputs as external knowledge, thereby improving its problem-solving capabilities over successive iterations.

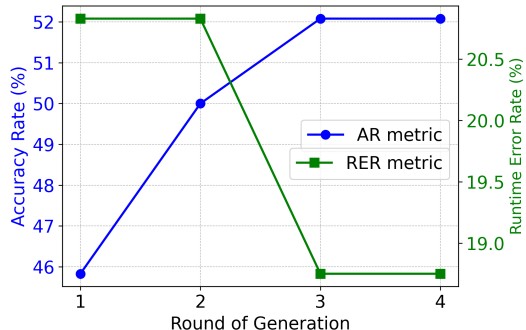

Figure 11: Performance of the DRoC-BBO.

Experimentally, the DRoC-BBO can slightly improve the performance with more rounds of generation with updated knowledge sources, which is shown in Figure 11. This indicates that the LLMs can also be enhanced through Bootstrap for solving optimization problems like VRP.

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
