# OpenReview forum: "DRoC: Elevating Large Language Models for Complex Vehicle Routing via Decomposed Retrieval of Constraints"
_ICLR.cc/2025/Conference — ICLR 2025 Poster_

### Official Review · Reviewer_sHTV · 2024-11-01

**Soundness:** 3
**Presentation:** 2
**Contribution:** 2
**Rating:** 6
**Confidence:** 4

**Summary:**

This work mainly established a novel RAG-LLM framework called DRoC which can resolve various complex VRPs with composite constraints. DRoC incorporates internal LLM knowledge and external information by decomposing the retrieval of constraints, deeply stimulating the intrinsic ability of LLM in intergrating different sources, and generating the problem-solving code without any training process. This approach can either flexibly apply vairous expert documentations or rigorously self-debug as circumstances may require, while iteratively optimizing the solution multiple times. The experimental results indicate that DRoC achieves SOTA preference in each metric compared with other homogeneous methods.

**Strengths:**

- This paper clearly states its research motivation and emphasizes why it is important to establish a novel LLM framework for solving complex VRPs.
- The methodology is rigorous and mature, paying attention to many practical details. This can be reflected in the mechanisms of the router, the debugger and the retriever with a refined structure.
- The experimental results in section 5 show a promising enhancement when compared to the listed former approaches. Ablations studies also support the structure design of DRoC.
- DRoC proposed a flexible problem-solving LLM-based framework without any training requirement. This framework could be referenced in other LLM scenarios with fast external information access.

**Weaknesses:**

- In my prospective, this is a pure prompt-based framework that may lack novelty and seem trivial. It would more significant if DRoC could shows more strong points other than SOTA preference, such as the generalization ability in analogous problems, remarkable inference efficiency, or convincing justification of framework structures, etc.
- The figure 2 can be more precise in order to guide the reader to better understand the framework structure. (1) The workflow of this flowchart is currently incomplete. For example, the "if not passed" desicion node needs "yes/no" paths to direct different outcomes. (2) The specific positions of each LLM module mentioned in appendix A should be indicated in this chart. For example, when does DRoC apply the first/second-stage filters?
- There is no specific dataset description in section 5. Appendix B provides an overall illustration of the considered VRP variants. However, the source of 48 problems remains unclear. This may affect the reproducibility of DRoC's experimental results.

**Questions:**

- It seems this is a code generation problem specifically. Is it more proper to add the keyword "code generation" to this paper?
- In figure 1, why are the numbers of different additional constraints not the same? A uniform standard is better needed, especially for the intro figure.
- At the end of section 4.2 "Single-constraint resolution", does "selecting the most relevant document" mean that currently DRoC retrieves information from only one document for each additional constraint? Is there any opportunity for LLMs to retrieve and digest information from multiple documents and make a composite utilizaiton of them?
- What are the standard prompts in section 5.1?
- What's the support for the statement at the end of section 5.4 "the LLMs can solve the 48 VRP variants with more composite constraints, indicating that complex tasks can be fulfilled by our decomposition-based method" ?
- I do not understand the motivations behind section 5.6 and 5.8, which seem not to be tightly related to the main topic of DRoC.

---

> ### Author Response · Authors · 2024-11-22
> **Response to Reviewer sHTV (1/3)**
>
> We sincerely appreciate the reviewer's acknowledgment of the strengths of our work, including the research motivation, rigorous methodolofy, and comprehensive experiment. We believe the paper is improved significantly after adopting the feedback from the reviewer, and we hope the responses below address the concerns effectively.
>
> **W1. It would more significant if DRoC could shows more strong points other than SOTA preference.**
>
> Thank you for highlighting this point. DRoC is designed as a generalizable framework capable of addressing problems with composite conditions, which aligns with the structure of many OR problems. To demonstrate its versatility, we have added examples in **Appendix F** of the revised manuscript, showcasing how DRoC successfully addresses the **assignment problem** and **facility location problems**—tasks where original LLMs show limited proficiency. For example, 9 variants of facility location problems are all successfully solved by gpt-4o+DRoC, while the standard gpt-4o only solves 5 of them. These results underscore DRoC’s potential as a plug-and-play enhancement for improving LLM performance in solving complex problems.
>
> As more and more online and open-source materials (solver tutorials and code snippets) emerge from the OR community, we believe our approach can take advantage of these resources that are already available to help LLMs solve complex optimization problems better.
>
> In addition to the generalizability to other OR problems, DRoC can also solve a class (variant) of the problem once the program is accurately generated. Unlike existing methods that prompt the LLM for each individual instance [1],  our method generates a reusable solving function, enabling the efficient solution of multiple instances within the same problem variant.
>
> **W2. Figure 2 is not precise enough.**
>
> We acknowledge that the figure can be improved for greater clarity and precision. In response to the reviewer’s feedback, we have updated the figure to provide a more comprehensive depiction of the workflow. Specifically, (1) we use the optional path to indicate some of the paths in the workflow are not necessary, and thus direct to different outcomes; (2) we incorporate all components of DRoC in the revised version.
>
> **W3. There is no specific dataset description.**
>
> Since there are currently no readily available datasets for the study of OR problems with composite constraints, we developed the dataset with the assistance of human experts by combining the common additional constraints. For each VRP variant, we use a simple instance for correctness verification, which ensures the feasibility of the solution and the efficiency of the unit test. We have added more explanations in **Appendix B** for clarification of the dataset. Please review the update highlighted in blue.
>
> We will also make the utilized dataset publicly available to facilitate the work's reproducibility, which can facilitate the research on LLM for solving complex OR problems.

---

> ### Author Response · Authors · 2024-11-22
> **Response to Reviewer sHTV (2/3)**
>
> **Q1. Is it more proper to add the keyword "code generation" to this paper?**
>
> We agree with the reviewer that this paper focuses on code generation for solving optimization tasks such as VRPs. In fact, this is explicitly stated at the beginning of Section 3.2: “We solve a code generation (or code completion) problem, without the mathematical model formulation…” The keyword “code generation” is also mentioned multiple times throughout the paper, reflecting its central role in our approach.
>
> As a work focuses on code generation, the novelty of our method compared to previous code generation techniques is that 1) we incorporate a decomposition method to generate code for complex real-world problems; 2) we integrate self-debug and RAG to simultaneously use the internal and external knowledge of the LLMs; 3) the method is shown to be effective for various OR problems. The experiments have showcased that our method significantly surpasses the typical techniques from the NLP coomunity (such as chain-of-thought and self-RAG) in code generation, and achieves better performance in complex VRPs.
>
> **Q2. Why are the numbers of different additional constraints not the same in Figure 1?**
>
> The variation in the number of additional constraints in Figure 1 arises from the way VRP variants are defined based on constraint combinations. For example, when the number of additional constraints ($w$) is 0, only basic problems like the Simple VRP and the Traveling Salesman Problem (TSP) are present, and there are no other simple routing problems like them. As constraints are combined, the number of possible VRP variants increases significantly, leading to more variants when $w \in [1,2,3]$.
>
> However, as w increases further, the number of designed variants decreases due to the **Incompatibility of Constraints**, which means that some constraints are inherently incompatible. For instance, the prize-collecting VRP already incorporates a distance limit and allows for dropping nodes, making certain additional constraints redundant (e.g., distance limit and open route) or infeasible. Therefore, the numbers of different additional constraints are not the same, depending on the problem variants of our dataset.
>
> **Q3. Does "selecting the most relevant document" mean that currently DRoC retrieves information from only one document for each additional constraint? Is there any opportunity for LLMs to retrieve and digest information from multiple documents and make a composite utilizaiton of them?**
>
> Thanks for the insightful suggestion. In the current implementation, DRoC retrieves one document per additional constraint, meaning that LLMs retrieve and digest information from multiple documents for each VRP with composite constraints.
>
> However, multiple documents for one constraint are also possible. Therefore, based on the suggestion, we explored the possibility of incorporating multiple documents (e.g., retrieving two documents per constraint) and conducted additional experiments. Empirically, we observed that this approach negatively impacted performance, with the Accuracy Rate (AR) decreasing from *45.83%* to *37.5%*. We think this decline can be explained in three aspects:
>
> **1) The same API/Modeling Processes**. The code for modeling each constraint is generally the same for a solver. For example, the API or modeling process required for a specific constraint is well-defined and structurally the same. Adding more documents, even if they correctly model the constraint, often results in redundancy rather than performance improvement.
>
> **2) Noisy documents**. External knowledge sources can contain a significant amount of noisy or irrelevant content. Including additional documents during the RAG process increases the risk of introducing irrelevant information, which can distract the LLM and degrade generation performance.
>
> **3) Long context**. Retrieving more documents expands the context length that the LLM needs to process. This makes it more challenging for the LLM to extract key information, leading to decreased performance due to information overload.
> While retrieving and utilizing multiple documents may hold potential in other tasks, our findings suggest that focusing on the most relevant document for each constraint is more effective for solving VRPs. We appreciate the reviewer’s suggestion and will continue exploring ways to refine the retrieval and utilization mechanisms to further enhance DRoC’s capability.
>
> While retrieving and utilizing multiple documents may hold potential in other tasks, our findings suggest that focusing on the most relevant document for each constraint is more effective for solving VRPs. We appreciate the reviewer’s suggestion and will continue exploring ways to refine the retrieval and utilization mechanisms to further enhance DRoC’s capability.

---

> ### Author Response · Authors · 2024-11-22
> **Response to Reviewer sHTV (3/3)**
>
> **Q4. What are the standard prompts in section 5.1?**
>
> It refers to directly prompting the LLMs to generate programs, which represents the most basic input-output solution for calling the LLMs. We have elaborated on the baseline approaches in **Appendix C** and provided its corresponding prompt.
>
> **Q5. What's the support for the statement at the end of section 5.4 ?**
>
> Thanks for highlighting this point. We have corroborated the statement in our experiments. As shown in Table 2,  without DRoC, gpt-4o can only accurately solve *2.08%* of the VRP variants. In contrast, with our decomposition-based DRoC framework, the success rate increases significantly to *31.25%* when using Gurobi, demonstrating that VRPs with more composite constraints can be successfully solved. This supports our statement that DRoC effectively enables LLMs to tackle more complex variants by leveraging decomposition. We also verified the same advantage by using another solver OR-Tools.
>
> However, we recognize that the original statement might have been unclear. To improve clarity, we have revised it to: “The LLMs can accurately solve more VRP variants with more than one constraint, indicating that complex tasks can be fulfilled by our decomposition-based method.” This revision better reflects the empirical findings and avoids potential confusion.
>
> **Q6. What are the motivations behind section 5.6 and 5.8?**
>
> The motivation for including Section 5.6 stems from the significance of LLM+EC methods, as highlighted in the literature review (Section 2.1). These methods represent a breakthrough in utilizing LLMs for solving OR problems [2][3]. Similar to our setting, their goal is to create a function for addressing a specific problem. However, while our work focuses on leveraging existing solvers, LLM+EC methods aim to evolve algorithmic templates.
>
> Given the overlap in objectives—*both leveraging LLMs to address problems such as VRPs*—we believe it is important to include a comparison and discussion of these two approaches. This allows stakeholders to better understand the strengths and limitations of each method, helping them decide which approach is best suited for specific problem-solving scenarios. The comparison results have shown our method is more efficient and competitive than EC methods, providing greater enhancement of the LLM.
>
> The motivation for Section 5.8, which *examines the benefit of dynamically updating external knowledge sources*, is closely tied to our method. While the main experiments utilize a static knowledge base, exploring the potential of a dynamic knowledge base is a natural extension of our approach. In this context, ‘dynamic’ refers to knowledge sources that can be updated by the LLMs themselves.
> This concept is intuitive because it raises the question of whether the accurate programs generated by the LLMs can be used to further enhance their own performance in a bootstrap manner. Including this discussion highlights an important point for enhancing DRoC in such optimization tasks.
>
> [1] Huang, Zhehui, Guangyao Shi, and Gaurav S. Sukhatme. "From Words to Routes: Applying Large Language Models to Vehicle Routing." arXiv preprint *arXiv:2403.10795* (2024).
>
> [2] Romera-Paredes, Bernardino, et al. "Mathematical discoveries from program search with large language models." *Nature* 625.7995 (2024): 468-475.
>
> [3] Liu, Fei, et al. "An example of evolutionary computation+ large language model beating human: Design of efficient guided local search." *International Conference on Machine Learning* (2024).

---

> > ### Comment · Reviewer_sHTV · 2024-11-23
> >
> > Thanks for your comprehensive responses. I appreciate your rigorous explanation to each question and my perpiexities have been well-resolved. It's a delight to help the improvement of this paper. I would like to raise my score after a careful consideration.

---

> > > ### Author Response · Authors · 2024-11-23
> > > **Thanks for the prompt feedback**
> > >
> > > We appreciate the reviewer for the prompt feedback and acknowledgement of our repsonses!

---

### Official Review · Reviewer_x6QC · 2024-11-03

**Soundness:** 2
**Presentation:** 3
**Contribution:** 2
**Rating:** 6
**Confidence:** 4

**Summary:**

The paper presents DRoC, an approach for LLM-based generation of VRP optimization models via decomposed retrieval of constraints. Specifically, DRoC iteratively improves solutions by using both self-debugging and specialized RAG that integrates external knowledge per constraint based on decomposition of the VRP constraints. Experiments on 48 VRP variants and two solvers (OR-Tools, Gurobi) show the proposed approach significantly outperforms the baselines.


----
Based on the authors' response and improvements to the paper, I have increased my evaluation.

**Strengths:**

Strengths:
- LLMs for operations research modelling is an important topic with significant recent interest.
- Experiments over 48 VRP variants show significant gains for the proposed approach
- Paper is mostly clear and written well.

**Weaknesses:**

Weaknesses:
- I have major concerns regarding the experimental evaluation:
	- Evaluation based on success rate can be misleading: the generated code could run successfully and lead to a feasible solution but not the correct one for the requirement. For example, it can ignore (relax) some of the constraints, thus keep the problem feasible.
	- Evaluation based on optimality gap is also unclear to me in this setting. Comparing optimality gaps makes sense when we compare the efficiency of solving or modelling techniques (assuming both candidates are solving the same problem). Here we are comparing the accuracy of the formulation. If you are not solving the same problem (e.g., because generation dropped a constraint or added a constraint while maintaining feasibility) it is not clear how this helps.
	- It is also not clear how the optimal solution (used in the OG computation) is obtained. Is it based on a ground-truth formulation, or on solving the generated instance for longer? While the appendix indicates a time limit of 100s to determine optimal solution, it is not clear what was the time limit for the solving of the generated instances.
	- Overall it seems like none of the evaluation actually talks about the correctness of the generated model.

- A more restricted setting (user provides both the name of the VRP variant and the signature of the function with clear documentation) compared to various previous work that supported natural language. This is a more restricted setting that requires more expertise from the user (both coding and familiarity with the literature). For example, if the user knows the problem they are solving is called "Capacitated Vehicle Routing Problem with Time Windows and Multiple Depots", they can probably decompose it themselves to the aspects of "capacitated" "time windows" and "multiple depots" (or that we could hard code those for each of the 48 problems, there are not that many) eliminating the need for a decomposer. This is not a flaw but it does restrict the potential impact of the approach.

- From a technical perspective there is limited innovation and it is more of a new LLM-based workflow. It is not entirely clear to me that these ideas will generalize well outside of VRP (even in the world of operations research), for example, the decomposer seems to rely heavily on the very useful naming of VRP problems that clearly define its constraints.

- Minor:
	* OG is presented as a fraction but presented in percentage.
	* According to OG formula it is not clear that OG is bounded at 1 (the difference can be larger than the optimal value), and therefore it is not clear why 1 is used for unsuccessful programs.

**Questions:**

I would appreciate if the authors could comment on the points mentioned above, in particular with respect to the experimental evaluation.

---

> ### Author Response · Authors · 2024-11-22
> **Response to Reviewer x6QC (1/3)**
>
> We sincerely thank the reviewer for the valuable feedback. We appreciate the recognition of the importance of LLMs for operations research modeling, as well as the acknowledgment of the significant gains demonstrated in our experiments. We are also glad that the clarity and quality of our paper were noted. Below, we provide our detailed responses to the points raised.
>
> **W1. Evaluation based on success rate and optimality gap is misleading.**
>
> We appreciate the reviewer’s observation regarding the limitations of the previous evaluation metrics. We realized that relying solely on the success rate (SR) can lead to the generation of feasible but incorrect solutions, so we combined the optimality gap (OG), which serves as an auxiliary indicator to estimate the actual solution quality. However, we agree that evaluating solely based on SR and OG can still be misleading, particularly as LLMs may relax constraints during code generation.
> In response to this valuable feedback, we have revised the evaluation framework. Inspired by prior research [1], we have replaced the original metrics with two robust alternatives: **Accuracy Rate (AR)** and **Runtime Error Rate (RER)**. We assess the solutions by using the new metrics to ensure more understandable and meaningful evaluations. These metrics are defined as follows:
>
> **Accuracy Rate (AR)**: $\text{AR} = \frac{V_a}{V_t} \times 100\%$, where$V_t$​ is the total number of generated programs for various VRP variants, and $V_a$​ represents the number of programs that successfully produce the optimal solution for a given variant. This metric directly reflects the correctness of the generated solutions.
>
> **Runtime Error Rate (RER)**: $\text{RER} = \frac{V_e}{V_t} \times 100\%$, where $V_e$ indicates the number of program that encounter runtime errors. RER measures the proportion of generated programs that encounter execution errors, such as internal logical issues caused by incorrect solver API calls or syntax errors.
>
> The AR metric evaluates the ability of LLMs to produce correct and optimal solutions, while RER captures their capacity to debug and generate feasible, error-free code. The AR metric is actually a more strict version of the SR used in the previous version. These revised metrics provide a more comprehensive and accurate assessment of the performance of our proposed method, DRoC, and the baselines. Accordingly, we have updated all the experiments in Section 5. For example, the current Table 2 is displayed below:
>
> | LLM                                    | AR ↑     | RER ↓     |
> |----------------------------------------|----------|-----------|
> | gpt-4o (Standard Prompting)          | 2.08%    | 45.83%    |
> | claude-3.5-sonnet (Standard Prompting)  | 12.50%   | 52.08%    |
> | gpt-4o (DRoC)                            | **31.25%** | **25.00%** |
> | claude-3.5-sonnet (DRoC)               | **31.25%** | **37.50%** |
>
> With the new metrics, we observed that the effectiveness of our method is more obvious, because accurately solving problems for the LLMs alone can be more difficult. For instance, the state-of-the-art gpt-4o achieves a success rate of only 2.08% when using Gurobi. In contrast, our proposed method, DRoC, demonstrates notable improvements in both AR and RER compared to all baselines. In summary, the updated results by the new metrics still highlight the advantages of our method and align with the findings from the original results, while making the experiments more reliable and convincing.
>
> **W2. It is not clear how the optimal solution (used in the OG computation) is obtained.**
>
> The optimal solution used in the OG computation is primarily obtained through Hybrid Genetic Search (HGS), a widely adopted method for generating ground truth solutions for VRP instances [2]. Given the small scale of the test instances, HGS can effectively produce optimal solutions, with the lowest objective values for feasible solutions.
>
> For implementation, we utilized the open-source HGS package provided by [3]. However, as the applicability of this implementation is limited, and it does not support VRPs with specific constraints of “Pickup and Delivery” or “Resource Constraint”, so OR-Tools was used to solve variants with those constraints. Given the relatively small scale of the instances, a search time limit of 100 seconds was set to find (near-)optimal solutions, which can be used to reflect the quality of solutions derived from DRoC and other baselines. In our updated paper, we elaborated more on the computation process in **Appendix B** for clarification.

---

> ### Author Response · Authors · 2024-11-22
> **Response to Reviewer x6QC (2/3)**
>
> **W3. A more restricted setting (user provides both the name of the VRP variant and the signature of the function with clear documentation) compared to various previous work that supported natural language. This is a more restricted setting that requires more expertise from the user (both coding and familiarity with the literature).**
>
> This paper actually focuses on a different perspective compared to previous works that supported natural language for solving OR problems. In particular, prior works often require a complete description of the problem, including background, scenario illustrations, coordinates, and node attributes for all customer nodes. While this can be effective for problems with short descriptions, LLMs frequently struggle with longer natural language contexts, losing coherence and accuracy in the process [4]. As the description length increases, the likelihood of generating correct solutions diminishes significantly, and this conclusion has already been verified by [5], which takes the TSP as a simple case.
>
> In contrast, this paper frames VRP solving as a **function completion task**, which enables the method to have the potential for handling larger-scale VRPs. By focusing on generating and verifying general-purpose functions, the approach ensures scalability and correctness. Furthermore, this method shifts the focus from *instance-based* problem-solving, as seen in prior works, to a *class-based* framework. DRoC generalizes solutions to an entire VRP class (a set of VRP instances), rather than target prompting or modeling each individual instance, which enhances its applicability.
>
> While this approach may require users to specify VRP variants and provide function signatures, it reduces reliance on verbose natural language descriptions and enables solving more complex and scalable VRP problems.
>
> Last but not least, DRoC, as a hybrid approach, leverages the powers of LLMs and well-developed solvers, aiming to tackle complex VRPs that are challenging for LLMs alone. DRoC is a promising trade-off to introducing little knowledge of VRP name and Python function (which are easy and friendly for users with simple backgrounds). Still, it significantly produces programs that are much better than LLM techniques in literature, as verified by our results.  This synergy allows DRoC to address a broader spectrum of VRPs more successfully and effectively.

---

> ### Author Response · Authors · 2024-11-22
> **Response to Reviewer x6QC (3/3)**
>
> **W4. From a technical perspective there is limited innovation and it is more of a new LLM-based workflow. It is not entirely clear to me that these ideas will generalize well outside of VRP (even in the world of operations research), for example, the decomposer seems to rely heavily on the very useful naming of VRP problems that clearly define its constraints.**
>
> While DRoC is presented in the context of VRPs, it represents a generalizable framework for addressing complex problems using a decomposition approach. This methodology can naturally extend to other OR problems, such as assignment, scheduling, and other constrained problems. For example, scheduling problems often involve additional constraints like machine intensity, set-up time, job precedence, time windows, job priorities, and processing variability. DRoC’s decomposition and filtering mechanisms can be adapted to tackle such constraints, leading to improved performance and scalability.
>
> To illustrate the generalizability of the method, we take two more types of OR problems (i.e., **Assignment** and **Facility location**) as examples, which are presented in **Appendix F**, where we also find our DRoC can help improve the performance of the LLMs. For instance, the results for the facility location problem are presented as below:
>
> |                                    | Contained in the external knowledge | Solved by gpt-3.5-turbo | Solved by gpt-4o | Solved by gpt-3.5-turbo (DRoC) | Solved by gpt-4o (DRoC) |
> |------------------------------------|-------------------------------------|-------------------------|------------------|--------------------------------|-------------------------|
> | $p$-median FLP                     | ✓                                   | ✘                       | ✓                | ✘                              | ✓                       |
> | $p$-center FLP                     | ✓                                   | ✘                       | ✓                | ✓                              | ✓                       |
> | $p$-maxian FLP                     | ✓                                   | ✘                       | ✓                | ✓                              | ✓                       |
> | $p$-dispersion FLP                 | ✓                                   | ✘                       | ✘                | ✓                              | ✓                       |
> | Capacitated FLP                    |                                     | ✘                       | ✓                | ✘                              | ✓                       |
> | Capacitated $p$-median FLP         |                                     | ✘                       | ✘                | ✓                              | ✓                       |
> | Capacitated $p$-center FLP         |                                     | ✘                       | ✓                | ✘                              | ✓                       |
> | Capacitated $p$-maxian FLP         | ✓                                   | ✘                       | ✘                | ✘                              | ✓                       |
> | Capacitated $p$-dispersion FLP     |                                     | ✘                       | ✘                | ✓                              | ✓                       |
>
> As the original gpt-3.5-turbo is unable to solve any of the 9 variants of the facility location problems, DRoC enables it to successfully solve 5 variants.
>
> While the current decomposer leverages the clear naming conventions of VRP variants, the framework itself is not limited to this structure. Its form can be adapted to suit other tasks, and task-specific decomposition can even be enhanced by fine-tuning a small LLM. This adaptability makes DRoC a promising approach for a wide range of applications beyond VRPs, in both OR and NLP domains. However, in this paper, we focus on solving VRPs with composite constraints, which are shown to be very challenging for LLMs. It can be more interesting to apply the framework of DRoC to other structured tasks in the future.
>
> **W5. OG is presented as a fraction but presented in percentage.**
>
> Thank you for pointing out the inconsistency. The representation of OG has been unified as a percentage throughout the paper for clarity and consistency.
>
> **W6. It is not clear why OG=1 is used for unsuccessful programs.**
>
> In our original paper, we set OG=1 because we aimed to increase the overall optimality gaps (for estimating the overall solution quality) if there are unsuccessful cases. Inspired by your feedback, in the revised evaluation, the original metrics have been replaced with the new metrics, AR and RER. With this change, the OG=1 setting has been removed. We believe the new metrics provide a more intuitive and accurate reflection of the performance of the generated programs. We have discussed the further advantages of the new metrics in our response to **W1**.

---

> ### Author Response · Authors · 2024-11-22
> **References**
>
> [1] Xiao, Ziyang, et al. "Chain-of-Experts: When LLMs Meet Complex Operations Research Problems." *The Twelfth International Conference on Learning Representations*. 2024.
>
> [2] Zhou, Jianan, et al. "Towards omni-generalizable neural methods for vehicle routing problems." *International Conference on Machine Learning*. PMLR, 2023.
>
> [3] Wouda, Niels A., Leon Lan, and Wouter Kool. "PyVRP: A high-performance VRP solver package." *INFORMS Journal on Computing* (2024).
>
> [4] Xu, Xin, et al. "Can LLMs Solve longer Math Word Problems Better?." arXiv preprint *arXiv:2405.14804* (2024)
>
> [5] Yang, Chengrun, et al. "Large Language Models as Optimizers." *The Twelfth International Conference on Learning Representations*. 2024.

---

> ### Author Response · Authors · 2024-11-28
>
> Dear Reviewer x6QC,
>
> We would like to sincerely thank you for your insightful and valuable feedback on our submission. We have carefully considered each of your points and made the necessary revisions accordingly.
>
> As the extended deadline for the discussion phase is approaching, we would be deeply appreciative if you could kindly review the changes we have made and let us know if there are any remaining concerns or suggestions. Thank you again for your time and effort in reviewing our work. We are truly grateful for your support and look forward to hearing from you soon.

---

> > ### Comment · Reviewer_x6QC · 2024-11-29
> >
> > Thank you for your response. I think the change in the evaluation metrics is important. It is still not entirely clear to me how the "Accuracy Rate" is computed: Do you check if that generated code is identical to a reference code (because there could be multiple programs that are "correct")?

---

> ### Author Response · Authors · 2024-11-29
> **Response to Reviewer x6QC**
>
> Thank you for the follow-up feedback. We appreciate the opportunity to clarify how we compute the "Accuracy Rate" (AR) in our evaluation.
>
> **Accuracy Rate**: AR is not determined by comparing the generated code to a reference implementation. This approach is impractical, as most problem variants do not have existing reference code. In fact, one of the key motivations for developing the DRoC framework is to assist users in quickly generating code for new problem variants with composite constraints, particularly when no prior implementations are available.While we developed reference implementations (mainly based on HGS) to generate optimal solutions for evaluation purposes, we emphasize that creating such implementations from scratch is time-consuming.
> Instead, in our AR computation, we check the correctness of the code in two steps.
>
> First, we consider a generated program accurate if it produces the same optimal objective value as a pre-computed optimal solution for the given problem instance. While being focused on similar tasks, we found this metric has been used in the literature [1], indicating its soundness. This approach evaluates correctness based on numerical outcomes rather than the specific code structure or implementation details. Since the objective values are float numbers, if the objective value of the generated program is the same as the optimal objective value, the generated program is supposed to be correct for the target problem. For example, when using OR-Tools, problems are solved by invoking specific APIs for specific constraints. If the generated code misuses these APIs, it is unlikely to achieve the optimal objective value, indicating inaccuracies in the implementation.
>
> Second, to verify that the generated programs are not only numerically correct but also logically sound, we also manually inspected them and ensured that they correctly involve all constraints and all the parameters are correctly used.
>
> **Multiple Correct Programs**: We acknowledge that there should be different implementations (i.e., codes) to solve the same problem, especially in optimization tasks where different modeling strategies can lead to the same optimal solution. For example, in solving VRPs by Gurobi, sub-tour elimination can be addressed by different strategies, such as using the enumeration technique or Miller-Tucker-Zemlin (MTZ) constraint. Both approaches are valid and deliver the same optimal objective value, even though the underlying code and formulations differ.
>
> Given the above point, we didn’t restrict LLM should only generate (or mimic) a fixed reference code, since multiple correct programs may exist. In our AR computation, we only check and ensure that the generated code correctly models the constraints and involves the parameters for the target problem, which can deliver the same optimal solution as the pre-computed optimal solution. In this manner, it allows for flexibility in generating any correct program for achieving optimal solutions, while maintaining AR as a **consistent and fair evaluation metric**.
>
> In summary, by using the optimal objective value as the criterion for accuracy, and further manual inspection on the programs, we effectively capture the correctness of the solution. Instead of mimicking a specific program, the AR reflects the true performance and maintains the flexibility of generating any correct program in solving optimization problems. The results with new metrics still showcase that DRoC framework is significantly superior to existing methods.
>
> [1] Xiao, Ziyang, et al. "Chain-of-Experts: When LLMs Meet Complex Operations Research Problems." The Twelfth International Conference on Learning Representations. 2024.

---

> ### Author Response · Authors · 2024-12-01
>
> Dear Reviewer x6QC:
>
> We would like to follow up on our recent discussion kindly. We greatly appreciated your question and the opportunity to address it in our previous responses.
>
> With the discussion deadline approaching in 2 days, we want to check if you have any additional feedback. Your input is valuable, and we would happily address anything further to ensure the work meets your expectations.
>
> Thank you for your time and thoughtful engagement. We look forward to hearing from you.

---

> ### Author Response · Authors · 2024-12-02
> **Follow-Up: Public Discussion Phase Ending Soon**
>
> Dear Reviewer x6QC,
>
> With the public discussion phase ending in less than 22 hours, we are eager to receive your feedback and confirmation on our rebuttal.
>
> For your convenience, we remind you by summarizing the revisions we've made to address all your concerns:
>
> **Evaluation:** By your suggestion, we have refined the evaluation metrics and re-conducted the experiments to ensure a fair assessment of program correctness. The updated results reaffirm the significant advantage of our method over the baselines.
>
> In response to your follow-up question, we clarified the soundness of the **Accuracy Rate** metric in our rebuttal and detailed how we ensure its reliability.
>
> **Generalization Beyond VRP:** We extended the proposed framework to evaluate its performance on other OR problems, demonstrating its broader applicability.
>
> **Minor Changes:** All minor concerns have been addressed and incorporated into the revised paper.
>
> Our rebuttal elaborates on these points in detail, and all corresponding revisions have been integrated into the paper. We are confident that we have comprehensively addressed your concerns.
>
> We kindly invite you to review the revisions before the discussion phase concludes. We appreciate your further feedback in advance!

---

### Official Review · Reviewer_45kN · 2024-11-04

**Soundness:** 3
**Presentation:** 3
**Contribution:** 2
**Rating:** 8
**Confidence:** 3

**Summary:**

The work proposes a framework for solving complex vehicle routing problems using LLMs. The framework's novelty is decomposed retrieval of constraints, where the VRP's constraints are decomposed into individual constraints, and then relevant conditioning documents are retrieved, and code generation is conditioned on those documents.

**Strengths:**

The  work combines an algorithmic solution with queries to LLM to solve challenging problems of operation research. The algorithms are described in detail, and accompanied by extensive empirical evaluation and ablation study.

**Weaknesses:**

Update: authors' clarifications and answers are quite convincing. Updating my evaluation accordingly.

Introduction: vehicle routing problems in computer science are not formulated or solved to route vehicle, contrary to what the introduction says. Many VRPs are hard problems to which other problems can be reduced to show hardness and devise approximations.

Methodology: including a code base on VRPs into the set of the documents for retrieval introduces data leakage and effectively replaces code generation with search for appropriate code. Integration with Gurobi, where only single constraint solutions are provided, does not convince otherwise because Gurobi is a specific solver where constraints are inherently and trivially composable.

Empirical evaluation: there are two important missing baselines without which the evaluation does not make much sense.
1. Any VRP problem can be solved optimally by enumeration. Judging by the figures, problem instances are small, so enumeration should be feasible if slow. What is the ratio of running times  by DRoC and by enumeration? On both successful and unsuccessful instances, mean and standard deviation of the log ratio of running times.

2.There are non-LLM based (including the code base used for conditioning) algorithm implementation for VRP problems used for evaluation (here is how the optimal solution metrics are obtained). Similarly, what is the ratio of running times (mean/standard deviation of log ratio)?

**Questions:**

What is the set of VRP problems the evaluation was performed on? Is this set publicly available?

---

> ### Author Response · Authors · 2024-11-22
> **Response to Reviewer 45kN (1/3)**
>
> We sincerely thank the reviewer for the valuable comments and appreciate the recognition of our detailed methodology description and extensive empirical evaluation. Below, we provide our detailed responses to the concerns raised.
>
> **W1. Vehicle routing problems in computer science are not formulated or solved to route vehicle, contrary to what the introduction says. Many VRPs are hard problems to which other problems can be reduced to show hardness and devise approximations.**
>
> We agree that in computer science, VRPs are often studied as canonical NP-hard problems to investigate computational complexity, approximation algorithms, and optimization techniques, rather than being directly tied to routing actual vehicles. Based on your comment, we have elaborated more on this point in the revised version of the paper (at the beginning of the introduction).
>
> However, we clarify that the primary focus of the paper is to automate the problem-solving process and enhance the practical applicability of VRPs in industrial scenarios. By bridging the gap between theoretical research and real-world applications in operations research (OR), our work aims to make VRP solvers more accessible and effective for handling various vehicle routing constraints (e.g., time windows, multiple depots, resource constraintss, etc.) from diverse practical use cases. Thus, in the revised paper, we start the introduction with the theoretical aspect and extend it to the application-oriented aspect.
>
> **W2. Including a code base on VRPs into the set of the documents for retrieval introduces data leakage and effectively replaces code generation with search for appropriate code.**
>
> Thanks for raising this concern. There may indeed be documents in the external knowledge that address similar problems to those DRoC is required to solve. If the external knowledge contains the same or similar code as the target solution, it is a feature, not a flaw of the approach, as it demonstrates the system's ability to retrieve and utilize accurate, pre-existing solutions to solve a problem. This mirrors real-world programming scenarios where developers often refer to online documentation, public repositories, or existing implementations to ensure correctness. Meanwhile, even if there is a similar code, the retrieved code cannot be executed directly without any modifications, based on the following reasons:
>
> **Incomplete Code Pieces**: The retrieved code files typically consist of partial, potentially useful snippets rather than fully usable programs. These snippets often represent step-by-step solutions tailored to specific problem instances. In contrast, our task focuses on completing a general-purpose solve function, which often differs significantly in structure and format from the existing examples.
>
> **Limited Scope of Existing Codes**: The community-contributed examples generally address VRPs with simple constraint structures. For instance, the OR-Tools online tutorial (https://developers.google.com/optimization/routing) provides code for individual constraints but lacks guidance on integrating multiple constraints effectively to solve more complex VRP variants. However, the solutions generated by DRoC are inherently different, going beyond the scope of the retrieved snippets to tackle more complex problems.
>
> Thus, DRoC is not a simple search system but a hybrid approach that combines code generation and retrieval. It leverages LLMs to generate code for simpler VRP variants, while relying on RAG to retrieve knowledge for handling complex constraints that are challenging for LLMs alone. This synergy allows DRoC to effectively address a broad spectrum of VRPs.

---

> ### Author Response · Authors · 2024-11-22
> **Response to Reviewer 45kN (2/3)**
>
> **W3. Integration with Gurobi, where only single constraint solutions are provided, does not convince otherwise because Gurobi is a specific solver where constraints are inherently and trivially composable.**
>
> While it's true that Gurobi, as a solver, can handle multiple constraints once they are formulated, the process of developing a program that accurately represents a VRP with multiple and complex constraints is non-trivial, and this is mainly because:
>
> **Constraint Interactions**: Each constraint may interact with others in non-linear and unexpected ways, requiring careful modeling to ensure the resulting formulation is valid and solvable. For instance, a prize-collecting TSP has a distinct optimization objective that allows skipping nodes, unlike other VRPs with single-constraint formulations where visiting all nodes is mandatory. These variations can lead to conflicts when integrating diverse constraints or objectives.
>
> **Impact of Core Constraints**: Typical constraints, such as flow conservation or sub-tour elimination, often influence the modeling of additional constraints. In some cases, these core constraints can render solutions infeasible. For example, sub-tour elimination may destroy all loops in the graph, making it impossible to generate feasible solutions once additional constraints are added. Handling such situations requires careful reasoning and debugging, often supported by retrieving relevant information to refine the model.
> Considering the above issues, we retrieve the single-constraint solutions and integrate the self-debug mechanism to resolve potential constraint interactions, making the LLMs model the VRPs more accurately.
>
> **W4. (1) Any VRP problem can be solved optimally by enumeration. Judging by the figures, problem instances are small, so enumeration should be feasible if slow. What is the ratio of running times by DRoC and by enumeration? On both successful and unsuccessful instances, mean and standard deviation of the log ratio of running times.**
>
> Thanks for the comment. While VRPs can theoretically be solved optimally by enumeration, our empirical analysis shows that this approach is infeasible for the instances in our study due to the exponential growth in complexity caused by composite constraints.
>
> For example, in the case of a Capacitated Vehicle Routing Problem (CVRP) with n customer nodes and m vehicles, each customer has m assignment options, leading to a total of $m^n \times n!$  possible solutions. For an instance with 17 nodes and 4 vehicles, this translates to an estimated computation time of approximately **26 years** by implementing the enumeration-based algorithm. To improve the efficiency of the enumeration-based method, we integrate advanced strategies in the numeration process: 1) route feasibility and distance caching; 2) early pruning of infeasible routes; 3) vectorized operations. After that, the computational time is reduced significantly, but we still need around **3000s** ($time_e$) to solve the CVRP instance. In comparison, the program generated by DRoC only took **less than 1s** ($time_d$) to give the optimal solution. Accordingly, the calculated log ratio ($log\frac{time_d}{time_e}$) is -8.0. Moreover, CVRP is one of the simpler variants in our study. Other VRPs involve additional composite constraints, which further exacerbate the complexity and hinder the practicality of enumeration.
>
> To sum up, it is very time-consuming to calculate the optimal solution through enumeration method, especially for the broader set of VRP variants addressed in this work. Instead, DRoC is designed to provide efficient, practical solutions for these complex optimization problems, making it well-suited for industrial and real-world applications where computational resources and time are limited.

---

> ### Author Response · Authors · 2024-11-22
> **Response to Reviewer 45kN (3/3)**
>
> **W4. (2) There are non-LLM based (including the code base used for conditioning) algorithm implementation for VRP problems used for evaluation (here is how the optimal solution metrics are obtained). Similarly, what is the ratio of running times (mean/standard deviation of log ratio)?**
>
> In our implementations, we used the Hybrid Genetic Search (HGS) solver for obtaining optimal solutions and OR-Tools in our RAG framework, respectively. Specifically, the running time for the HGS solver is supposed to be configured, which serves as the stopping condition for the search process. We set a maximum run time of 1 second for HGS, as we found it is sufficient to obtain optimal solutions for the simple instances used in our evaluation. For the OR-Tools solver, we set the search time limit as 100s, as it has an early stop mechanism once there is no solution improvement. The statistics for running time are as:
>
> | Method            | Mean Time (s) | Std Time (s) |  Mean Log Ratio | Std Log Ratio|
> |---------------------|--------------------|------------------|-----------------------|-------------------|
> | HGS                |        1.005       |         0.003    |          -                |            -          |
> | DRoC (HGS)   |        0.329       |       0.713      |         -3.257        |       0.317       |
> | OR-Tools         |        12.503     |      32.379     |          -                |            -          |
> | DRoC (OR-Tools) |    1.411      |       4.059      |         -1.145        |       0.882       |
>
> Note that we calculate the running time of DRoC for instances solvable by HGS [denoted as DRoC (HGS)] and those solvable by OR-Tools [denoted as DRoC (OR-Tools)], respectively.
> The results show that the mean and standard deviation of the running time for DRoC (OR-Tools) are smaller than those of directly calling OR-Tools. This is mainly because the search limit parameter is not configured for most of the instances in the programs generated by DRoC, while we predetermine the search limit for OR-Tools when calculating the approximated optimal solutions. When we set the same time limit for both OR-Tools and DRoC (OR-Tools), the time metrics can be nearly identical (with a mean log ratio of 0.004). This is intuitive, as the generated code produced by DRoC maintains a similar structure to manually written code for solving specific VRPs. Thus, DRoC does not introduce any significant increase in running time, while it provides an automatic and user-friendly approach to solve the problems.
>
> **Q1. What is the set of VRP problems the evaluation was performed on? Is this set publicly available?**
>
> Thanks for the question. As there is no publicly available dataset encompassing the wide variety of VRP variants, we developed the dataset ourselves with the assistance of human experts in the OR domain. We summarize the common additional constraints (as shown in section 3.1) and combine them to create the VRP variants. We used a simple instance for each VRP variant to create test cases. This approach ensured that each problem instance was feasible for testing the solution function. To provide greater transparency, we have added a more detailed description in **Appendix B**. Additionally, we are preparing the dataset for public release to support reproducibility and future research in this domain.

---

> ### Author Response · Authors · 2024-11-28
>
> Dear Reviewer 45kN,
>
> We would like to sincerely thank you for your thoughtful and valuable feedback on our submission. We have carefully addressed each of your points and made revisions accordingly. Given that the extended discussion deadline is approaching, we would greatly appreciate it if you could review the changes and let us know if there are any remaining concerns.
>
> Once again, thank you for your time and support in reviewing our work. We look forward to hearing from you soon.

---

> ### Author Response · Authors · 2024-12-01
> **Your Feedback Would Be Appreciated**
>
> Dear Reviewer 45kN:
>
> We want to kindly follow up on the feedback we requested earlier regarding our paper revisions. With the discussion deadline approaching in 2 days, we are hoping to understand your perspective or address any concerns you may have.
>
> We deeply value the time and effort you dedicate to the review process. Thank you so much for your consideration.

---

> > ### Comment · Reviewer_45kN · 2024-12-01
> >
> > Thank you for the clarifications, they address most of the issues I raised. I would appreciate the clarifications finding their way into the final version of the paper, if accepted.

---

> ### Author Response · Authors · 2024-12-01
>
> Dear Reviewer 45kN:
>
> Thank you for increasing the score and supporting the acceptance of our paper!
>
> We're pleased to know that the clarifications addressed the issues you raised. We assure you that all corresponding revisions will be incorporated into the final version of the paper. We sincerely appreciate your time and thoughtful feedback!

---

### Official Review · Reviewer_i3uy · 2024-11-09

**Soundness:** 3
**Presentation:** 3
**Contribution:** 3
**Rating:** 6
**Confidence:** 3

**Summary:**

This work is along the popular direction of using LLM to automatically generate modeling code and then call solvers in the backend to solve the problem. Specifically, it proposes a novel framework that enhances LLMs in code generation to solve complex vehicle routing problems (VRPs) with intricate constraints. Traditional LLMs perform well on simpler VRPs but struggle with complex versions due to a lack of embedded domain-specific knowledge. The proposed DRoC framework addresses this by using a retrieval-augmented generation (RAG) approach to incorporate external knowledge for constraint modeling.

**Strengths:**

The problem is well motivated. It is important to be solve different variants of VRP problems. From the numerical results, it appears that with the proposed approach, the accuracy of modeling is improved substantially.

**Weaknesses:**

1. The RAG technique is common for code generation. What is the challenge of applying RAG in solving VRP? The novelty of this paper is limited.
2. In practice, there are also other variants of VRP which cannot be solved immediately by the solvers. The paper does not address the case where the problem at hand cannot be solved by the VRP solver.

**Questions:**

1. What happens if the given VRP variant cannot be solved by the solver?
2. In the experiment, the authors use "optimality gap" as a criterion. Is the feasibility verified first? Otherwise we would have objective value that is not feasible, or even lower than the optimal value of that problem.

---

> ### Author Response · Authors · 2024-11-22
> **Response to Reviewer i3uy (1/2)**
>
> We would like to thank the reviewer for the valuable comments and for recognizing the strong motivation behind our method, as well as its superior performance. We believe the feedback from the reviewer has improved the paper significantly. We will address the concerns raised below.
>
> **W1. The RAG technique is common for code generation. What is the challenge of applying RAG in solving VRP? The novelty of this paper is limited.**
>
> We acknowledge that some approaches used RAG in code generation. However, applying RAG to complex optimization problems like VRPs presents significant challenges:
>
> **Complex Constraint Structures**: VRPs involve diverse and interdependent constraints, where changes in one constraint often affect others. The previous RAG approaches rarely account for this high interdependency and struggle to address composite constraints. As shown in Table 1, the vanilla RAG approach (direct retrieval of problem-related documents) performs poorly and sometimes degrades performance.
>
> **Granularity of Retrieval**: VRP solvers like OR-Tools and Gurobi provide documentation or examples for broad problem types (e.g., CVRP, TSP) but lack specificity for complex variants (e.g., CVRPTW with multiple depots). Retrieving such generalized examples may lead to missing key details specific to the required constraints, and this is one reason that the performance of vanilla RAG is not satisfactory.
>
> **Numerous and Noisy Documentation**: VRP-related documentation and community-contributed examples are extensive but often irrelevant. For example, if we try to retrieve (using vanilla RAG) text chunks from the online tutorial of OR-Tools by the keyword “Traveling Salesperson Problem”, we only get the first sentence on the page (https://developers.google.com/optimization/routing/tsp): “This section presents an example that shows how to solve the Traveling Salesperson Problem (TSP) for the locations shown on the map below”, which is not helpful for code generation.
>
> To address these issues, DRoC innovatively incorporates domain-specific decomposition and filtering mechanisms tailored to optimization problems like VRP, which is different from the existing RAG-based code generation tasks.
>
> On the other hand, this is the first time we apply RAG to solve VRP with complex constraints. While some existing work studies simple TSP and CVRP, we employ RAG to focus more on correctness of modelling complex VRP, making DRoC a unified framework for various VRPs. Our experiments have corroborated that our RAG framework, with novel decomposition and filtering designs, is much better than traditional RAG techniques in solving complex VRPs..
>
> **W2. In practice, there are also other variants of VRP which cannot be solved immediately by the solvers. The paper does not address the case where the problem at hand cannot be solved by the VRP solver.**
>
> Thanks for raising this insightful point. We recognize that certain VRP variants may pose additional challenges that cannot be directly solved using standard VRP solvers. However, most VRP variants can be expressed as Mixed-Integer Programming (MILP) problems, which mature solvers like Gurobi or CPLEX are well-equipped to handle. In our experiments, we have corroborated that our method can be easily employed with Gurobi to solve more VRP variants.
>
> In addition, metaheuristics provide a flexible alternative to addressing unsolvable variants, which can be also used in our RAG framework. For example, an LLM could assess whether a given variant can be solved by the available VRP solver and, if not, guide the creation of suitable metaheuristics. In the process of metaheuristic creation, RAG can play a key role by retrieving relevant documentation to assist the LLM in making informed decisions and improving constraint-wise components of the metaheuristic. In this paper, we focused on solvers since they already covered many common constraints, which also avoid the creation of metaheuristics from scratch.
>
> Finally, we note that this paper aims to enhance the solver-calling capabilities of LLMs and help users leverage existing solvers more effectively, rather than addressing (uncommon) VRPs that cannot be solved by these solvers. Compared to existing LLM-based VRP solutions, the proposed method already covers a much broader range of VRPs, especially covering complex variants. Addressing more problem types is an interesting research topic.  In the future, we will extend LLMs to seamlessly integrate solver selection and metaheuristic generation to further enhance their versatility and reliability in addressing more diverse optimization tasks. We have added this future work to our conclusion.

---

> ### Author Response · Authors · 2024-11-22
> **Response to Reviewer i3uy (2/2)**
>
> **Q1. What happens if the given VRP variant cannot be solved by the solver?**
>
> If the VRP variant cannot be solved by the selected solver, it may result in an unsuccessful outcome, such as a compilation error during the execution of the generated program. To address this, as noted earlier, a practical solution is to empower LLMs to generate metaheuristics when standard solvers are inadequate. This approach would enable the LLM to adapt dynamically to problem-specific requirements, ensuring robust problem-solving even for unconventional VRP variants, which however introduces more inference time of LLMs.
>
> However, we emphasize that the primary target of this paper is to **enhance the usability and user-friendliness of existing solvers** by reducing human effort. The proposed method focuses on effectively leveraging these solvers for solving a wide range of VRP variants. Notably, the approach already addresses more complex VRP constraints than those typically addressed in the literature, which often focus only on solving TSP and simple VRPs using neural networks or LLMs, demonstrating significant progress in this area.
>
> **Q2. In the experiment, the authors use "optimality gap" as a criterion. Is the feasibility verified first? Otherwise we would have objective value that is not feasible, or even lower than the optimal value of that problem.**
>
> Thanks for raising this important point. In our original paper, we designed a success rate (SC) and optimality gap (OG) for evaluation. While there could be infeasible solutions recognized as successful, we further use the optimality gap to indicate the gap between the generated solution and the optimal solution. The quality of the produced solution can be reflected by referring to these two metrics together. For example, if a solution is successful but infeasible, it may cause a high gap (signified by 1) for the OG metric which is substantially larger than the optimal solution.
>
> However, to make the evaluation more rigorous and understandable, we have replaced the original metrics (SC and OG) with *Accuracy Rate (AR)* and *Runtime Error Rate (RER)*. AR is a more strict version of SC, which measures whether the objective value of the generated solution matches the optimal solution, ensuring that feasibility is inherently verified and satisfied. RER tracks the occurrence of runtime errors, capturing the reliability of the generated solutions.  These new evaluation metrics ensure that we avoid situations where the LLM generates infeasible solutions or achieves lower-than-optimal objectives by relaxing constraints. We have reevaluated different methods using new metrics. **The results in the paper, including Table 1, Table 2, Table 3, Figure 1, Figure 3, Figure 5, and Figure 10, are all updated.** The updated results still highlight the advantages of our method and align with the findings from the original results, while making the experiments more reliable and convincing.

---

> ### Author Response · Authors · 2024-11-28
>
> Dear Reviewer i3uy,
>
> We sincerely appreciate your detailed feedback on our manuscript. We have carefully addressed each of your comments and made corresponding revisions.
>
> As the revised discussion phase deadline is approaching, we wiil appreciate it very much if you let us know if you have any further concerns or feedback, and we are eager to ensure that all issues are resolved in our submission. Thank you again for your valuable insights and support.

---

> ### Author Response · Authors · 2024-12-01
> **Your Feedback Would Be Appreciated**
>
> Dear Reviewer i3uy:
>
> We want to kindly follow up on the feedback we requested earlier regarding our paper revisions. With the discussion deadline approaching in 2 days, we are hoping to understand your perspective or address any concerns you may have.
>
> We deeply value the time and effort you dedicate to the review process. Thank you so much for your consideration.

---

> ### Author Response · Authors · 2024-12-02
> **ICLR Public Discussion Phase Ends in One Day**
>
> Dear Reviewer i3uy,
>
> As the public discussion phase is set to close in less than 22 hours, we want to ensure that all your questions and concerns have been fully addressed. We have carefully reviewed your feedback and provided detailed responses to each of your comments. We sincerely hope our clarifications and revisions meet your expectations and resolve any outstanding issues.
>
> Thank you for your valuable feedback. We look forward to hearing from you!

---

> > ### Comment · Reviewer_i3uy · 2024-12-03
> >
> > I have read the authors feedback. I'll keep my score.

---

> > > ### Author Response · Authors · 2024-12-03
> > > **Thank you for your review**
> > >
> > > Dear Reviewer i3uy,
> > >
> > > Thank you for taking the time to review our paper. We greatly appreciate your recognition of our work, particularly regarding its motivation and superior performance. Your thoughtful feedback is very valuable to us!

---

### Author Response · Authors · 2024-11-25
**General Response to the Area Chair and Reviewers**

We would like to sincerely thank all reviewers for their thorough evaluation, insightful comments, and constructive suggestions on our work. We deeply appreciate the recognition of the strengths of our proposed DRoC framework, including its motivation, rigorous methodology, and its SOTA performance on complex VRP variants.The valuable feedback has significantly helped us improve the clarity, presentation, and depth of our submission.

In response to the main concerns raised, we have undertaken the following actions to further enhance our submission:

**1. Improved Evaluation Metrics:** To address concerns about the previous evaluation metrics, we have replaced the success rate (SR) and optimality gap (OG) with more robust metrics: Accuracy Rate (AR) and Runtime Error Rate (RER). These new metrics provide a clearer and stricter assessment of the correctness and reliability of generated solutions. The results reaffirm the effectiveness of DRoC in outperforming all baseline methods. (Corresponding revisions in the paper: Table 1- Table 3, Figure 1,3,5,10, and discussions in Section 5.)

**2. Generalization and Broader Impact:** We have extended the discussion of DRoC’s generalizability to analogous operations research problems, including assignment and facility location problems, where we again observe that our method works well. This demonstrates the potential of DRoC as a plug-and-play solution for other structured optimization problems beyond VRPs, indicating the potential of its broader applicability. (Corresponding revisions in the paper: Table 5, Figure 11, and Appendix F.)

**3. Enhanced Clarity and Transparency:** We have revised our introduction to reflect the theoretical and application-oriented aspects more accurately. Additionally, we have provided more detailed descriptions of the dataset and evaluation setup, ensuring greater reproducibility of our results. We also updated figures and additional explanations to further explain workflow clarity and dataset representation. (Corresponding revisions in the paper: Section 1, Appendix B, and Figure 2.)

**4. Discussion of Running Time:** We have conducted additional comparisons and provided evidence that our method does not significantly improve the solver running time by using generated programs. It highlights the practical advantages of DRoC in terms of efficiency, performance, and applicability to the real world. (Corresponding revisions in the paper: Appendix G and Table 7.)

**5. Addressing Specific Weaknesses:** We have directly responded to all reviewer comments regarding novelty, evaluation rigor, and technical details. We further clarified the motivations behind specific sections and potential future work, addressed concerns about potential data leakage, and elaborated on how DRoC balances retrieval and generation to effectively handle composite constraints. (Corresponding revisions in the paper:  Appendix B and Section 6.)

We hope the detailed responses and updated submission reflect our commitment to addressing the concerns and demonstrate the significance of our contributions to the field. Thank you again for your time and effort in reviewing our work. We look forward to any further feedback and are happy to provide additional clarifications as needed.

Sincerely,

Authors of Submission9519

---

### Meta-Review · Area_Chair_c4VA · 2024-12-22

**Metareview:**

The paper aims to address the vehicle routing problems (VRP), a complex constrained optimization problem, using large language models. It is based on the retrieval augmented generation framework, by integrating external knowledge and decomposition techniques to enhance LLMs in program generation. Existing VRP solvers are integrated in this loop to enhance LLM capabilities. The writing of the paper is of good quality, and the evaluation is comprehensive (with many baselines) and convincing. The main weakness of the paper is its limited novelty, but reviewers reached a consensus that this paper is comparatively better than other similar work on LLM applications and the problem it addresses is important. Thus I am inclined to accept this paper.

**Additional Comments On Reviewer Discussion:**

Reviewers and authors had good discussions during the discussion period, and all technical concerns and questions were addressed. Reviewers are generally ok with the quality of the paper and have no objection to accepting this paper.

---

### Decision · Program_Chairs · 2025-01-22

Accept (Poster)